# GTPO and GRPO-S:
# Token and Sequence-Level Reward Shaping with Policy Entropy

Hongze Tan [* 1 2]  Zihan Wang [* 1 3]  Jianfei Pan [* 1]  Jinghao Lin [* 1 3]  Hao Wang [* 1 4]
Yifan Wu [* 5]  Tao Chen [1]  Zhihang Zheng [1]  Zhihao Tang [† 1 6]  Haihua Yang [♠ † 1]

## Abstract

Reinforcement Learning (RL) is pivotal for enhancing Large Language Model (LLM) reasoning, yet mainstream algorithms such as GRPO and DAPO remain constrained by a coarse-grained credit assignment paradigm, where all tokens within the same response receive the identical reward. In this paper, we propose **Dynamic Entropy Weighting**, systematically define entropy-based weight ratios $\frac{H_{i,t}}{\sum_{k=1}^{n} H_{k,t}}$ and similar variants to redistribute rewards and get fine-grained rewards through two new algorithms: **Group Token Policy Optimization (GTPO)**, which assigns an entropy-weighted reward to each token and synthesizes token-specific advantage function to drive the model toward optimal path, and the analogous algorithm **Sequence-Level GRPO (GRPO-S)**, which extends this design to the sequence level and exhibits superior stability in long Chain-of-Thought (CoT) reasoning tasks. Unlike methods using entropy as mere regularization, GTPO and GRPO-S establish a new state-of-the-art on AIME and MATH 500, outperforming prior entropy-guided baselines and validating our weighting mechanism.

## 1. Introduction

Large-scale Reinforcement Learning (RL) propels LLMs from pattern recognition to complex reasoning, enabling dynamic, multi-step problem-solving (Guo et al., 2025; Wei et al., 2022; Zhang et al., 2022; Achiam et al., 2023; Kimi et al., 2025; Jin et al., 2025). Standard RL approaches optimize policies by maximizing cumulative rewards, typically relying on a value function to estimate

expected future returns. However, due to the computational burden of maintaining such critic models, the field evolves from resource-intensive PPO-based pipelines (Mnih et al., 2015; Amini et al., 2024; Christiano et al., 2017; OpenAI Spinning Up, 2018; Gao et al., 2023) to streamlined, value-function-free paradigms. Exemplified by methods such as GRPO, this approach bypasses explicit critic models by leveraging outcome-based group baselines, thereby reducing the computational overhead for massive models (Shao et al., 2024; Yu et al., 2025; Feng et al., 2025b;a).

Despite the remarkable performance of algorithms like GRPO in mathematical and long Chain-of-Thought (CoT) reasoning tasks, a critical bottleneck remains: **coarse-grained credit assignment** (Uesato et al., 2022; Wei et al., 2022). Algorithms like GRPO assign a uniform reward to every token in a sequence based solely on the final outcome (Shao et al., 2024). This limitation becomes profound in tasks requiring long-chain reasoning (Cui et al., 2025a; Rafailov et al., 2024). For instance, a sequence with dozens of correct logical steps may receive zero reward for a single final error, penalizing correct and incorrect reasoning alike (Yang et al., 2025; Wang et al., 2025b). Conversely, a sequence that reaches a correct answer through flawed or guessed intermediate steps is fully rewarded (Sutton & Barto, 2018; Zhang et al., 2025).

We identify policy entropy as an intrinsic anchor for fine-grained credit assignment, distinguishing our work from recent approaches that treat it merely as a regularization constraint (Cui et al., 2025b) or a binary selection mask (Wang et al., 2025b). While these methods recognize that high entropy signals pivotal reasoning steps (Cheng et al., 2025), they perform passive filtering and fail to address the bottleneck of coarse-grained rewards. In contrast, our framework repurposes entropy for active reward shaping. Instead of discarding tokens, we dynamically redistribute the reward signal. By reinforcing high-entropy exploration in correct solutions while penalizing confident errors in incorrect ones, we achieve precise token-level supervision without an explicit value network.

The central observation is that moments of high policy entropy within a reasoning sequence are not random noise

---

[1]ByteDance China  [2]HKUST  [3]Northeastern University  [4]CityUHK  [5]SUSTech  [6]BUPT  [*]Equal Contribution  [†]Corresponding Authors  [♠]Project Leader .

*Proceedings of the 43rd International Conference on Machine Learning*, Seoul, South Korea. PMLR 306, 2026. Copyright 2026 by the author(s).

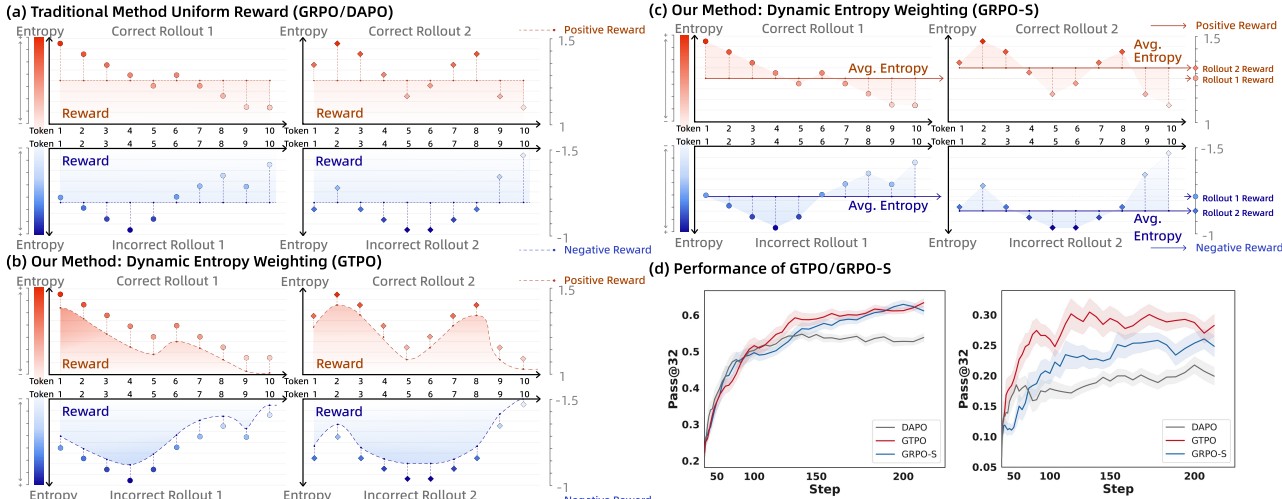

**Figure 1.** Conceptual illustration of reward assignment. (a) Previous methods assign a uniform reward based on the final outcome. In contrast, our methods use Dynamic Entropy Weighting to refine credit assignment: (b) GTPO rewards high-entropy tokens in correct sequences while suppressing them in incorrect ones, and (c) GRPO-S rewards correct sequences with higher average entropy while penalizing incorrect paths, yielding superior performance on AIME 2024 (left) and AIME 2025 (right) (d).

but strong correlates of pivotal reasoning junctures (Cheng et al., 2025; Cui et al., 2025b). When a model selects between multiple valid mathematical theorems or constructs a complex logical connective, its uncertainty, as measured by entropy, naturally increases. This policy entropy, viewed as a measure of model indecision, can be repurposed as a powerful heuristic for cognitive effort (Haarnoja et al., 2018; Lindsay, 2020). In successful paths, it signals a moment of valuable exploration to be reinforced; in unsuccessful paths, it can help the policy break from incorrect thinking. This principle motivates the proposal of **Dynamic Entropy Weighting**, a novel framework that reshapes the reward signal to be proportional to token-level or sequence-level entropy, thereby focusing the policy gradient on the most critical decision points.

We operationalize this framework through a suite of two complementary algorithms. The first, **Group Token Policy Optimization (GTPO)**, is a novel token-level algorithm that assigns a unique, entropy-weighted reward to every token, achieving the first true, fine-grained, per-token credit assignment within the efficient GRPO framework. Complementing this, **Sequence-Level Group Relative Policy Optimization (GRPO-S)** is a lightweight variant that modulates the global reward for an entire sequence based on its average entropy. Together, these methods offer a principled trade-off between micro-level granular precision and macro-level sequence stability, as illustrated in Fig. 1, which conceptually demonstrates how entropy modulation at both token and sequence levels leads to more nuanced credit assignment and improved performance.

This paper makes the following principal contributions:

- We demonstrate that GRPO still performs coarse-

grained credit assignment, and motivated by this, we fundamentally optimize the GRPO framework.

- We propose **Group Token Policy Optimization (GTPO)**, a novel token-level algorithm that introduces a dynamic, entropy-weighted reward mechanism to achieve precise, per-token credit assignment within the efficient GRPO framework.

- We develop **Sequence-Level Group Relative Policy Optimization (GRPO-S)** as an analogous sequence-level algorithm that uses the dynamic, entropy-weighted reward mechanism to capture the exploratory value of an entire sequence.

- We provide a theoretical analysis of unbiasedness and convergence to motivate our objective functions design and conduct comprehensive experiments on challenging reasoning benchmarks, showing that our methods significantly outperform strong baselines.

## 2. Dynamic Entropy Weighting For Policy Optimization

We now present the Dynamic Entropy Weighting framework in detail. Motivated by the statistical limitations of GRPO (§ 2.1), we present the detailed formulations of GTPO (§ 2.2) and GRPO-S (§ 2.3) with a theoretical analysis regarding the unbiasedness and convergence (§ 2.4).

### 2.1. From Coarse-Grained Credit Assignment to Dynamic Entropy Weighting

**Background: Group Relative Policy Optimization and Its Limitations.** Our work builds upon the Group Relative

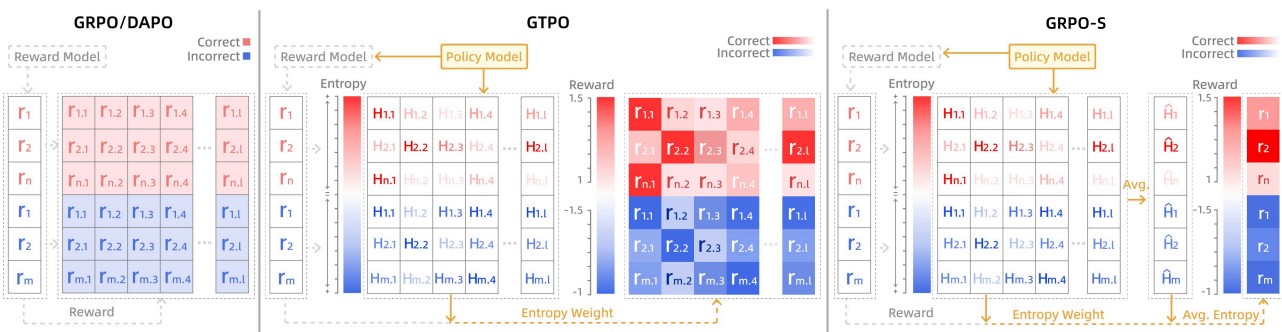

**Figure 2.** A high-level comparison of the reward signaling process. Methods such as GRPO and DAPO use a static reward model to assign a uniform reward to an entire sequence. Our framework, encompassing GTPO and GRPO-S, introduces a Dynamic Entropy Weighting module that reshapes this signal into fine-grained rewards at either the token or sequence level before it is used by the policy model.

Policy Optimization (GRPO) framework (Shao et al., 2024), a value-function-free algorithm that simplifies policy optimization for LLMs. Given a prompt $q$, GRPO samples a group of $G$ sequences, $\{o_1, o_2, \ldots, o_G\}$, from a policy $\pi_\theta$. Each sequence $o_i$ receives a terminal reward $r_i$ (e.g., 1 for correct, 0 for incorrect). The advantage function for all tokens within a sequence $o_i$ is defined as the sequence's reward normalized relative to the group's average reward:

$$\hat{A}_{i,t} = \hat{A}_i = \frac{r_i - \text{mean}(\{r_k\}_{k=1}^G)}{\text{std}(\{r_k\}_{k=1}^G)}. \quad (1)$$

The GRPO objective then applies this uniform advantage estimate to every token in the sequence within a PPO-style clipped loss function:

$$\mathcal{J}_{\text{GRPO}}(\theta) = \mathbb{E}\left[\frac{1}{G}\sum_{i=1}^G \frac{1}{|o_i|}\sum_{t=1}^{|o_i|}\min\left(w_{i,t}(\theta)\hat{A}_{i,t}, \text{clip}(w_{i,t}(\theta), 1-\epsilon, 1+\epsilon)\hat{A}_{i,t}\right)\right]. \quad (2)$$

While simple and effective, this uniform application of the advantage is the central mechanism of GRPO, and it also represents the core limitation that motivates our work: **coarse-grained credit assignment**. This approach is not only conceptually imprecise but, as will now be demonstrated, also statistically suboptimal.

**Motivation: The Statistical Case for Finer-Grained Advantage Estimation.** A key motivation for our shift towards a token-level objective stems from a variance reduction argument concerning the baseline term in the advantage function (i.e., the group's average reward). When sequence lengths $|o_i|$ are unequal, there are two primary ways to estimate this baseline:

**Sequence-level mean:** $\quad \hat{R}_1 = \frac{1}{G}\sum_{i=1}^G r_i.$

**Token-level mean:** $\quad \hat{R}_2 = \frac{\sum_{i=1}^G |o_i|r_i}{\sum_{i=1}^G |o_i|}.$

While a token-level reward baseline provably reduces variance for more stable gradients ( $\text{Var}(\hat{R}_2) \leq \text{Var}(\hat{R}_1)$,

see Appendix B.1 for a formal proof ). It is easy to see that the design of GRPO essentially estimates reward by **Sequence-level mean** $\hat{R}_1$, but according to the analysis in Appendix B.1 & B.2, we know that **Token-level mean** $\hat{R}_2$ is better. However, this exposes a critical issue: GRPO exhibits a dependency on the length of the response sequence. Specifically, for a correct response, it implies that "**the longer**, **the better**", which is obviously incorrect. Therefore, in a sense, GRPO still performs coarse-grained credit assignment. The underlying cause of this issue is that the design of the reward signal leads to an advantage function that fails to capture the specificity of each token.

**Solution: The Dynamic Entropy Weighting Framework.** To fully exploit this granularity, a principled reshaping of the reward signal itself is essential. This is achieved through dynamic entropy weighting, which, as illustrated in Fig. 2, focuses the policy gradient on critical decision points to create a far more instructive and fine-grained learning signal than uniform credit assignment approaches.

The framework partitions sequences (in $O$, where $O$ is the set consists of all response sequences) by their terminal reward into **successful** ($O^+$) and **unsuccessful** ($O^-$) sets, enabling a dual strategy for credit assignment. High-entropy tokens in **successful sequence** ($o_i \in O^+$) receive a reward bonus to reinforce valuable exploration. Conversely, low-entropy tokens in **unsuccessful sequence** ($o_j \in O^-$) are assigned relatively larger penalties to discourage confident but incorrect reasoning. This precise modulation focuses the policy gradient on the most informative steps.

### 2.2. Group Token Policy Optimization (GTPO)

Group Token Policy Optimization (GTPO) is the most direct and granular implementation of our framework. It introduces a fine-grained, entropy-weighted credit assignment mechanism that operates at the token level.

**Token-Level Reward Shaping.** For any token $o_{i,t}$ within a **successful sequence** $o_i \in O^+$, $o_{j,t}$ within an **unsuccessful**

**sequence** $o_j \in O^-$, the following entropy-weighted reward is defined:

$$\tilde{r}_{i,t}^+ = \alpha_1 r_i + \alpha_2 \frac{H_{i,t}}{\sum_{k=1}^n H_{k,t}} \cdot d_t, \qquad \tilde{r}_{j,t}^+ = 0. \quad (3)$$

This reward is composed of the original binary success signal $r_i$ (where $r_i = 1$) and a dynamic entropy bonus, balanced by hyperparameters $\alpha_1, \alpha_2 > 0$. The bonus is proportional to the token's generation entropy

$$H_{i,t} = -\sum_{v \in \mathcal{V}} \pi_{\theta_{\text{old}}}(v|q, o_{i,<t}) \log \pi_{\theta_{\text{old}}}(v|q, o_{i,<t}). \quad (4)$$

Crucially, this entropy is normalized across all $n$ successful sequences at timestep $t$, creating a *relative* signal that rewards valuable exploration, specifically targeting tokens generated with higher uncertainty compared to alternative successful paths (if a sequence $o_i$ has length less than $t$, its $H_{k,t}$ is treated as 0). This relative bonus is then scaled by $d_t$, the count of successful sequences with length $\geq t$, which dynamically adjusts the reward magnitude to account for the diminishing number of active reasoning paths over time.

For any token $o_{j,t}$ within an **unsuccessful sequence** $o_j \in O^-$, the goal is to penalize confident mistakes more heavily. The reward $\tilde{r}_{j,t}^-$ is thus defined using inverse entropy to assign relatively larger penalties to low-entropy (i.e., high-confidence) tokens:

$$\tilde{r}_{j,t}^- = \alpha_1 \cdot (-1) + \alpha_2 \frac{1/H_{j,t}}{\sum_{k=1}^m (1/H_{n+k,t})} \cdot h_t \cdot (-1),$$
$$\tilde{r}_{i,t}^- = 0. \quad (5)$$

where $h_t$ is the count of unsuccessful sequences with length $\geq t$ (if a sequence $o_j$ has length less than $t$, its $1/H_{j,t}$ is treated as 0). This formulation encourages the model to be uncertain when it is incorrect, promoting exploration away from failure modes. Based on these shaped rewards, separate advantage functions are computed for the positive and negative sets, both normalized over all tokens in the entire batch to ensure a consistent scale:

$$\tilde{A}_{i,t}^+ = \frac{\tilde{r}_{i,t}^+ - \text{mean}(\tilde{R}^+)}{\text{std}(\tilde{R}^+)} \quad \text{and} \quad \tilde{A}_{j,t}^- = \frac{\tilde{r}_{j,t}^- - \text{mean}(\tilde{R}^-)}{\text{std}(\tilde{R}^-)}. \quad (6)$$

Here, $\tilde{R}^+$ and $\tilde{R}^-$ represent the collections of all shaped token rewards across all positive and negative sequences in the group, respectively.

**The GTPO Objective Function.** The final objective function for GTPO integrates these components into a unified, token-level PPO-style loss. The expectation is taken over all tokens in the batch, weighted by the reciprocal of the total number of tokens, $1/\sum_{k=1}^G |o_k|$:

$$\mathcal{J}_{\text{GTPO}}(\theta) = \mathbb{E}\Bigg[\frac{1}{\sum_{k=1}^G |o_k|}\bigg(\sum_{i=1}^n \sum_{t=1}^{|o_i|} \min\left(w_{i,t}(\theta)\tilde{A}_{i,t}^+, \text{clip}(w_{i,t}(\theta), 1-\epsilon, 1+\epsilon)\tilde{A}_{i,t}^+\right)$$
$$+ \sum_{j=n+1}^G \sum_{t=1}^{|o_j|} \min\left(w_{j,t}(\theta)\tilde{A}_{j,t}^-, \text{clip}(w_{j,t}(\theta), 1-\epsilon, 1+\epsilon)\tilde{A}_{j,t}^-\right)\bigg)\Bigg], \quad (7)$$

where $w_{i,t}(\theta) = \frac{\pi_\theta(o_{i,t}|q, o_{i,<t})}{\pi_{\theta_{\text{old}}}(o_{i,t}|q, o_{i,<t})}$ is the standard importance sampling weight.

## 2.3. A Sequence-Level Variant of GTPO (GRPO-S)

While GTPO offers maximal granularity, it incurs computational overhead for per-token entropy and reward calculation. Additionally, since some tasks are result-oriented, the goal is to develop a corresponding sequence-level algorithm by following the approach of GTPO, and perform further refinement. Hence GRPO-S is proposed as an analogous method that applies our entropy-weighting principle at the sequence level. The core idea is to modulate the reward for an entire sequence based on its overall exploratory value, as captured by its average entropy.

**Sequence-Level Reward Shaping.** For any sequence $o_k \in O$, rewards are shaped based on a **sequence's average token entropy**,

$$\hat{H}_k = \frac{1}{|o_k|} \sum_{t=1}^{|o_k|} H_{k,t}. \quad (8)$$

For any successful sequence $o_i \in O^+$, the reward is augmented with an entropy-based bonus to reinforce valuable exploration. Conversely, for any unsuccessful sequence $o_j \in O^-$, an additional penalty proportional is applied to its average inverse entropy, thus penalizing high-confidence mistakes more severely. Formally:

$$\hat{r}_i^+ = \beta_1 r_i + \beta_2 \frac{\hat{H}_i}{\sum_{k=1}^n \hat{H}_k} \cdot n,$$
$$\hat{r}_j^- = \beta_1 \cdot (-1) + \beta_2 \frac{1/\hat{H}_j}{\sum_{k=1}^m (1/\hat{H}_{n+k})} \cdot m \cdot (-1). \quad (9)$$

where $\beta_1, \beta_2 > 0$ are hyperparameters and $\hat{r}_i^- = \hat{r}_j^+ = 0$. This formulation rewards successful sequences that are, on average, more exploratory, while penalizing confidently incorrect sequences.

**The GRPO-S Objective Function.** The advantage functions $\hat{A}_i^+$ and $\hat{A}_j^-$ are computed analogously to Eq. (1), but using the sequence-level shaped rewards and normalizing over the $G$ sequences in the group. The final objective function for GRPO-S mirrors the structure of the

original GRPO loss, but with our shaped advantages:

$$\mathcal{J}_{\text{GRPO-S}}(\theta) = \mathbb{E}\Big[\frac{1}{G}\Big(\sum_{i=1}^{n} \min\big(\hat{w}_i(\theta)\hat{A}_i^{+}, \text{clip}(\hat{w}_i(\theta), 1-\epsilon, 1+\epsilon)\hat{A}_i^{+}\big)$$
$$+ \sum_{j=n+1}^{G} \min\big(\hat{w}_j(\theta)\hat{A}_j^{-}, \text{clip}(\hat{w}_j(\theta), 1-\epsilon, 1+\epsilon)\hat{A}_j^{-}\big)\Big)\Big],$$

(10)

where $\hat{w}_i(\theta)$ is the sequence-level importance weight that averages the token-level weights:

$$\hat{w}_i(\theta) = \frac{1}{|o_i|}\sum_{t=1}^{|o_i|} w_{i,t}(\theta) = \frac{1}{|o_i|}\sum_{t=1}^{|o_i|} \frac{\pi_\theta(o_{i,t}|q, o_{i,<t})}{\pi_{\theta_{\text{old}}}(o_{i,t}|q, o_{i,<t})}.$$

(11)

*Remark* 2.1. Note that the definitions of our new algorithms involve multiple fractional terms. In practical implementation, a small positive constant $\epsilon$ needs to be added where necessary to ensure numerical stability. For the sake of brevity and to avoid cluttering the definitions, we omit explicit notation of this constant in the text.

## 2.4. Implementation and Theoretical Guarantees

### 2.4.1. IMPLEMENTATION DETAILS

This section consolidates the practical details required to implement our framework, including a critical mechanism for ensuring theoretical guarantees and a detailed algorithmic procedure.

**Algorithmic Procedure.** To clearly illustrate the implementation flow of GTPO and GRPO-S, the complete training procedure is provided in Algorithm 1. The procedure highlights the shared steps and the key differences between the token-level and sequence-level approaches.

### 2.4.2. THEORETICAL GUARANTEES

We provide a theoretical analysis establishing the unbiasedness and convergence of our framework, using GTPO as a representative case. Detailed proofs and a parallel analysis for GRPO-S are deferred to Appendices B.3 and B.4.

**Proposition 2.2** (Conservation of Positive Reward Mass - GTPO). *Let $O_t^{+}$ be the set of successful sequences active at timestep $t$ within a sampled group, with cardinality $d_t = |O_t^{+}|$. Under the GTPO reward shaping mechanism defined in Eq.(3), if the hyperparameters satisfy $\alpha_1 + \alpha_2 = 1$, then the sum of the shaped positive rewards equals the sum of the original binary rewards at every timestep:*

$$\sum_{i \in O_t^{+}} \tilde{r}_{i,t}^{+} = \sum_{i \in O_t^{+}} r_i.$$

(12)

**Proposition 2.3** (Asymptotic Consistency - GTPO). *Let $\Delta\mathcal{J}(\theta) = \nabla\mathcal{J}_{GTPO}^{+}(\theta) - \nabla\mathcal{J}_{GRPO}(\theta)$ be the gradient*

---

**Algorithm 1** GTPO and GRPO-S Training Procedure

**Input:** Initial policy model $\pi_\theta$; Task prompts $\mathcal{D}$; Hyperparameters $\alpha, \beta, G, \epsilon$; Learning rate $\eta$

1: **Initialize:** $\pi_{\theta_{\text{old}}} \leftarrow \pi_\theta$
2: **for** iteration $k = 1, 2, \dots$ **do**
    **Step 1: Data Collection & Entropy Calculation**
3:    $\{o_1, \dots, o_G\} \sim \pi_{\theta_{\text{old}}}(\cdot|q)$     # Sample $G$ outputs
4:    $r_i \leftarrow R(o_i) \in \{0, 1\}, \forall i$     # Get terminal rewards
5:    $O^{+}, O^{-} \leftarrow \text{Partition}(O)$    # Split based on outcome
6:    $H_{i,t} \leftarrow -\sum \pi_{\theta_{\text{old}}} \log \pi_{\theta_{\text{old}}}$     # Token-level entropy

    **Step 2: Reward Shaping & Optimization**
7:    **if Method** is GTPO (Token-Level) **then**
8:      $\tilde{r}_{i,t}^{+}, \tilde{r}_{j,t}^{-} \leftarrow$ Eq. (3), (5)   # Shape rewards via $H_{i,t}$
9:      $\tilde{A}_{*,t}^{+;-} \leftarrow \tilde{r}_{*,t}^{+;-}$        # Token Advantage (Eq. 6)
10:    $\mathcal{L} \leftarrow \mathcal{J}_{\text{GTPO}}(\theta; \tilde{A}_{*,t}^{+;-})$       # Loss via Eq. (7)
11:    **else if Method** is GRPO-S (Sequence-Level) **then**
12:      $\hat{H}_i \leftarrow \text{Avg}(H_{i,t})$          # Sequence entropy
13:      $\hat{r}_i^{+}, \hat{r}_j^{-} \leftarrow$ Eq. (9)      # Shape rewards via $\hat{H}_i$
14:      $\hat{A}_{*}^{+,-} \leftarrow \hat{r}_{*}^{+,-}$         # Sequence Advantage
15:      $\hat{w}_i \leftarrow$ Eq. (11)     # Sequence importance weight
16:      $\mathcal{L} \leftarrow \mathcal{J}_{\text{GRPO-S}}(\theta; \hat{A}_i, \hat{w}_i)$    # Loss via Eq. (10)
17:    **end if**
18:    $\theta \leftarrow \theta + \eta\nabla_\theta\mathcal{L}$           # Gradient Update
19:    $\pi_{\theta_{\text{old}}} \leftarrow \pi_\theta$               # Sync policy
20: **end for**
**Output:** $\pi_{\theta_{\text{old}}}$

---

*bias introduced by the entropy-weighted reward shaping, where $\mathcal{J}_{GTPO}^{+}(\theta)$ is the positive reward component of GTPO. Assume the policy $\pi_\theta$ satisfies the Entropy Consolidation Condition: as training iteration $k \to \infty$, the variation in token entropy among successful sequences diminishes, i.e., for any two successful sequences $o_{i_1}, o_{i_2} \in O^{+}$, $\lim_{k\to\infty} \frac{H_{i_1,t}}{H_{i_2,t}} = 1$ (almost surely, assuming regularization $\epsilon > 0$ prevents singularity). Then, the gradient bias vanishes asymptotically:*

$$\lim_{k\to\infty} \|\Delta\mathcal{J}(\theta_k)\| = 0.$$

(13)

**Theorem 2.4** (The Same Global Optimum - GTPO). *GTPO shares the same global optimum as DAPO.*

*Proof.* Proposition 2.2 establishes that the positive reward mass of GTPO is identical to that of GRPO. It is worth noting that, from a mathematical perspective, the design of DAPO is nearly identical to GRPO, with the primary distinction being the assignment of a reward of -1 to incorrect answers rather than the 0 reward used in GRPO. Consequently, following a derivation analogous to Proposition 2.1, we can conclude that the total reward mass of GTPO is equivalent to that of DAPO. The conservation of

**Table 1.** Comparison of different methods on various benchmarks.

| | AIME 2024 | | | AIME 2025 | | | MATH 500 |
|---|---|---|---|---|---|---|---|
| **Method** | **Mean@32** | **Pass@4** | **Pass@32** | **Mean@32** | **Pass@4** | **Pass@32** | **Mean@32** |
| | | | *Qwen2.5-7B* | | | | |
| GRPO | 28.36 | 36.15 | 46.01 | 15.00 | 19.03 | 20.00 | 78.62 |
| DAPO | 21.2 | 30.04 | 35.51 | 13.33 | 16.28 | 16.67 | 78.85 |
| DAPO w/ Forking Tokens | 30.77 | 41.34 | 54.16 | 15.21 | 19.82 | 21.67 | 79.25 |
| GTPO | 34.23 | **48.03** | **64.89** | 16.67 | **25.51** | **26.67** | **80.16** |
| GRPO-S | **34.67** | 47.18 | 64.76 | **18.33** | 22.76 | 23.33 | 80.07 |
| | | | *Qwen2.5-32B* | | | | |
| GRPO | 29.17 | 37.18 | 46.30 | 17.71 | 25.95 | 28.85 | 84.16 |
| DAPO | 34.06 | 46.64 | 59.02 | 21.67 | 25.76 | 26.67 | 84.45 |
| DAPO w/ Forking Tokens | 34.68 | 47.81 | 64.12 | 23.19 | 27.99 | 32.02 | 85.21 |
| GTPO | 35.21 | 49.41 | **68.91** | **26.89** | **33.49** | **36.67** | 85.72 |
| GRPO-S | **35.52** | **51.12** | 67.19 | 25.11 | 29.75 | 34.33 | **85.79** |

reward mass implies that the GTPO objective is an unbiased augmentation of the DAPO objective in terms of the total learning signal.

Note that the gradient estimator for the positive reward component of GTPO can be conceptually decomposed into:

$$\mathbb{E}[\nabla \mathcal{J}^+_{\text{GTPO}}] = \mathbb{E}[\nabla \mathcal{J}_{\text{GRPO}}] + \underbrace{\mathbb{E}\left[\text{Cov}\left(\Delta_{\text{entropy}}, \nabla \log \pi_\theta\right)\right]}_{\text{Exploration Term}},$$
(14)

where $\Delta_{\text{entropy}}$ represents the zero-sum redistribution of rewards based on uncertainty. Since the total positive reward remains tied to the ground truth $r_i$, the model cannot maximize the objective solely by manipulating entropy; it must eventually increase the likelihood of correct answers to gain more total reward mass. Furthermore, as $k \to \infty$, the entropy term $H_{i_1,t}, H_{i_2,t} \to 0$ and become uniform across successful paths, causing $\frac{H_{i_1,t}}{H_{i_2,t}} \to 1$ and $\tilde{r}^+_{i,t} \to r_i$. Based on Proposition 2.3, we know that the positive reward component of GTPO shares the same global optimum as GRPO. By extension, given that DAPO corresponds to GRPO subject to a simple reward shift for incorrect answers that preserves the optimal policy, which is a modification consistent with our own reward design for incorrect answers, it follows that GTPO shares the same global optimum as DAPO. $\square$

*Remark* 2.5. Our analysis focuses on the reward redistribution mechanism, which generates a granular, token-aware learning signal. This shaping is sign-consistent with the original objective, amplifying positive outcomes while penalizing negative ones. Furthermore, as the entropy term is detached from the gradient computation, it reshapes the reward landscape to favor exploration. Consequently, our analysis confirms that while these modifications influence training dynamics, they ensure optimization towards a valid policy optimum.

*Remark* 2.6. Note that while Proposition 2.2 assumes $\alpha_1 + \alpha_2 = 1$, this condition is not strictly necessary. The proofs remain valid even if $\alpha_1 + \alpha_2 \neq 1$, as the term $\alpha_1 + \alpha_2$ merely acts as a constant scaling factor that does not affect the validity of the derivation. In our experimental setup, the sum $\alpha_1 + \alpha_2$ is configured to be close to 1, but strict equality is not enforced.

*Remark* 2.7. The theoretical analysis of GRPO-S is analogous. See Appendix B.4.

## 3. Experiments

### 3.1. Experimental Setup

**Tasks and Datasets.** Methods are evaluated on the **AIME 2024**, **AIME 2025**, and **MATH 500** benchmarks (Hendrycks et al., 2021; Lightman et al., 2023). These challenging mathematical datasets require long-horizon thinking(Cobbe et al., 2021), making them a rigorous testbed for assessing advanced reasoning capabilities.

**Evaluation Metrics.** Our primary metric is **Pass@k** ($k \in \{2, 4, 8, 16, 32\}$), which encourages solution diversity over the more conservative Pass@1. **Mean@32** is also reported. Improvements are quantified by **Absolute (APG)** and **Relative (RPG) Performance Gains** over the baseline.

**Models and Baselines.** Experiments are conducted on **Qwen2.5-7B/32B** (Qwen et al., 2025) models. Comparisons are made against faithful implementations of **GRPO** (Shao et al., 2024) and **DAPO** (Yu et al., 2025), as well as the advanced **DAPO w/ Forking Tokens** (Wang et al., 2025a), to ensure a comprehensive evaluation of baselines.

**Implementation Details.** Experiments are run on 64 GPUs with a global batch size of 128, a group size of 16, and a learning rate of $1 \times 10^{-6}$. For generation, a temperature 1.0 and top-p 1.0 are used, with max lengths of 2048 (prompt)

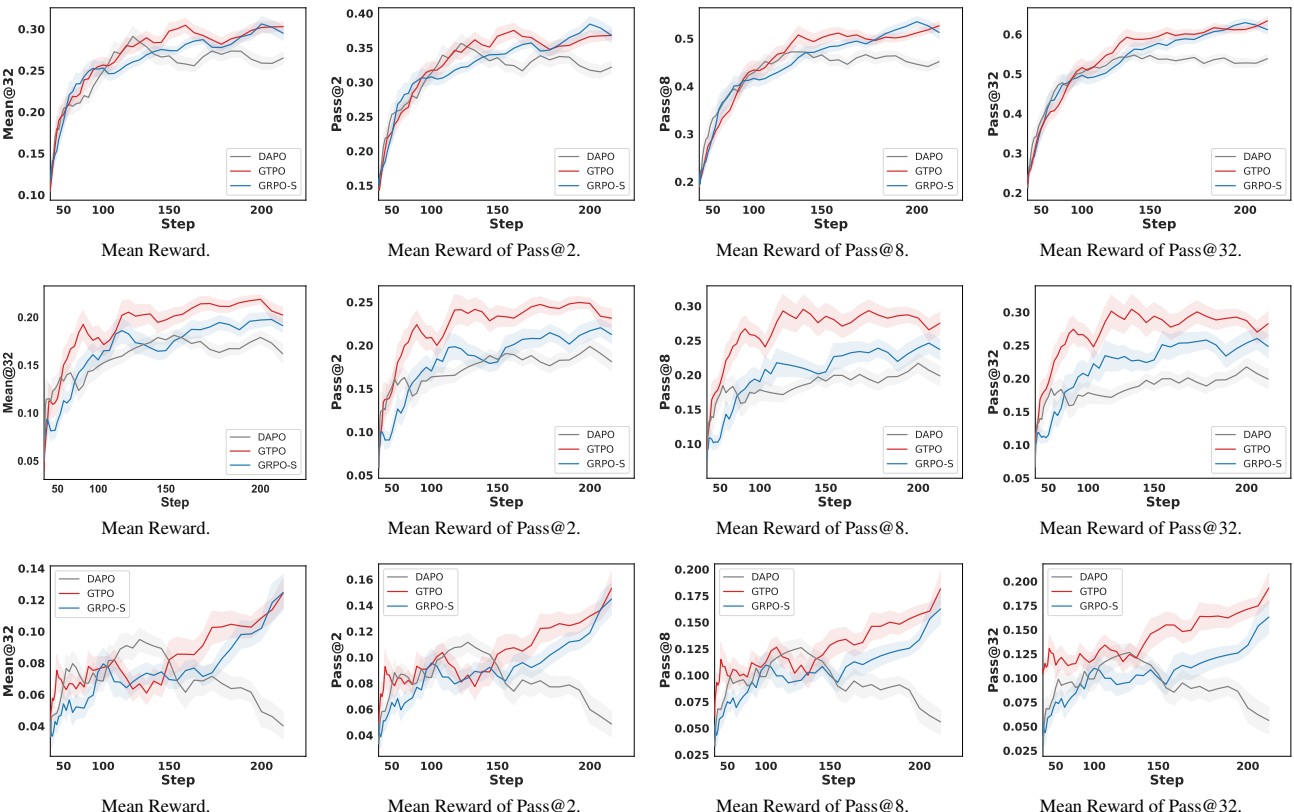

**Figure 3.** Mean reward trajectories on the test sets. **All curves are smoothed for visual clarity.** Each row corresponds to a different experimental setting: (Top) AIME 2024 with Qwen2.5-32B, (Middle) AIME 2025 with Qwen2.5-32B, and (Bottom) AIME 2025 with Qwen2.5-7B. Columns show different metrics from left to right: Mean Reward, Mean Reward of Pass@2, Pass@8, and Pass@32. For brevity and to maintain visual clarity, the corresponding results for Pass@4 and Pass@16, which exhibit similar trends, are deferred to Appendix E.3, see Fig. 8 and Fig. 9.

and 4096 (response). Key reward shaping hyperparameters are $\alpha_1 = \beta_1 = 1$, $\alpha_2 = \beta_2 = 0.1$, and entropy is clipped at $\epsilon_{low} = 0.2, \epsilon_{high} = 0.28$. Furthermore, to mitigate length bias, we employ the Group Relative Overlong Punishment mechanism detailed in Appendix D, which adaptively penalizes excessive token lengths based on the classification of problem difficulty (e.g., easy vs. hard).

### 3.2. Comparative Performance Analysis

As presented in Table 1, our methods, GTPO and GRPO-S, establish a new state-of-the-art by consistently outperforming GRPO, DAPO, and the advanced DAPO w/ Forking Tokens across all configurations, including the rigorous MATH 500 benchmark. Notably, our approach fundamentally differs from DAPO w/ Forking Tokens, which rigidly discards low-entropy tokens via a binary mask. Instead, we employ a continuous entropy-based reward shaping mechanism that fully leverages the learning signal across the distribution. This strategy successfully resolves the stability-exploration trade-off: while DAPO achieves stability at the cost of exploration (plateauing on challenging tasks like AIME 2025), our methods

significantly elevate exploration-sensitive scores. This efficacy is most pronounced on smaller models; specifically, regarding the Pass@32 metric on AIME 2024, GTPO achieves a massive absolute performance gain (APG) of **+29.4 points** on the 7B model, compared to **+9.9 points** on the 32B model, confirming that entropy-weighting provides a vital learning signal to prevent premature convergence.

### 3.3. Reward Trajectories and Sample Efficiency

The mean reward trajectories on the test set (Fig. 3) illuminate the benefits of our approach, demonstrating that our methods not only achieve a significantly higher final reward ceiling but do so with strong sample efficiency. The models largely converge within **210 training steps**, a finding substantiated by the training set reward curves, which are deferred to Appendix E.4 for brevity. This rapid convergence indicates that the benefits of enhanced exploration do not come at the cost of slower learning.

Our methods foster sustained exploration and prevent policy collapse, establishing a clear causal chain from our entropy-weighted reward to superior performance. A detailed analysis and the empirical evidence for this mechanism

Method 1 (Column-wise):

$$\begin{bmatrix} H_{1,1} & \ldots & \color{red}{H_{1,t}} & \ldots & H_{1,l} \\ H_{2,1} & \ldots & \color{red}{H_{2,t}} & \ldots & H_{2,l} \\ \vdots & \ddots & \vdots & \ddots & \vdots \\ H_{i,1} & \ldots & \color{green}{H_{i,t}} & \ldots & H_{i,l} \\ \vdots & \ddots & \vdots & \ddots & \vdots \\ H_{n,1} & \ldots & \color{red}{H_{n,t}} & \ldots & H_{n,l} \end{bmatrix}$$

Method 2 (Row-wise):

$$\begin{bmatrix} H_{1,1} & \ldots & H_{1,t} & \ldots & H_{1,l} \\ H_{2,1} & \ldots & H_{2,t} & \ldots & H_{2,l} \\ \vdots & \ddots & \vdots & \ddots & \vdots \\ \color{red}{H_{i,1}} & \ldots & \color{green}{H_{i,t}} & \ldots & \color{red}{H_{i,l}} \\ \vdots & \ddots & \vdots & \ddots & \vdots \\ H_{n,1} & \ldots & H_{n,t} & \ldots & H_{n,l} \end{bmatrix}$$

Method 3 (Matrix-wise):

$$\begin{bmatrix} \color{red}{H_{1,1}} & \ldots & \color{red}{H_{1,t}} & \ldots & \color{red}{H_{1,l}} \\ \color{red}{H_{2,1}} & \ldots & \color{red}{H_{2,t}} & \ldots & \color{red}{H_{2,l}} \\ \vdots & \ddots & \vdots & \ddots & \vdots \\ \color{red}{H_{i,1}} & \ldots & \color{green}{H_{i,t}} & \ldots & \color{red}{H_{i,l}} \\ \vdots & \ddots & \vdots & \ddots & \vdots \\ \color{red}{H_{n,1}} & \ldots & \color{red}{H_{n,t}} & \ldots & \color{red}{H_{n,l}} \end{bmatrix}$$

**Figure 4.** Comparison of three different methods.

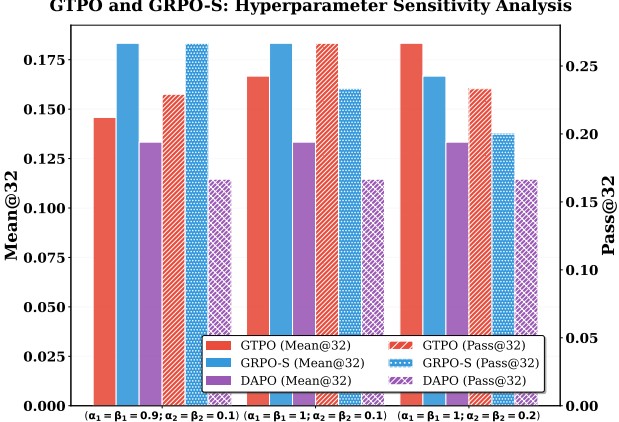

**Figure 5.** Hyperparameter comparison.

are presented in Appendix E.1.

### 3.4. Hyperparameter Sensitivity Analysis

Reward shaping adds extra degrees of freedom that control the trade-off between exploration and optimizing the primary objective. To test this balance between exploration and exploitation. We conduct a sensitivity analysis on the key reward shaping hyperparameters for GTPO and GRPO-S, with results presented in Fig. 5. Across all tested configurations, both of our methods demonstrate robust and significant performance gains over the DAPO baseline on Mean@32 and Pass@32 metrics. Among them, GRPO-S exhibits higher stability across settings, providing a clear view of the performance dynamics. As the weight of the entropy bonus ($\beta_2$) increases from 0.1 to 0.2, we observe a clear decline in overall performance for GRPO-S. This finding confirms that while exploration is beneficial, an excessive entropy bonus can detract from optimizing the primary task objective.

### 4. Discussion

**Batch-Level Entropy Comparison as Implicit Curriculum Learning.** Our current implementation operates at the batch level, which compares entropy across different problems, creating an efficient implicit curriculum. The algorithm naturally directs larger gradients towards solvable but high-entropy problems that represent the

frontier of the model's capabilities. As the model gains proficiency, the entropy of these problems decreases, causing the learning focus to automatically shift to the next set of challenging tasks. This design leverages the model's own uncertainty as a dynamic signal for learning priority, thus avoiding the need for curriculum design.

**Future Work.** The concept of relative entropy itself warrants deeper exploration. While this work highlights its importance, the optimal method for comparison remains an open question. Three distinct approaches for measuring this relativity are identified. Visualizing the token entropies as a matrix $H$, where $H_{i,t}$ is the entropy of the $t$-th token in the $i$-th response, these methods can be understood as: a **column-wise comparison**, which compares the entropy of tokens at the same position across different responses (i.e., within a column of $H$); a **row-wise comparison**, which compares tokens at different positions within the same response (i.e., within a row of $H$); and a **matrix-wise comparison**, which compares all tokens from all responses collectively (i.e., across the entire matrix $H$). Further experimentation is needed to verify which of these approaches is superior. The conceptual difference is illustrated in Fig. 4.

### 5. Conclusion

In this paper, we address the fundamental challenge of coarse-grained credit assignment, a critical flaw in aligning large language models for complex reasoning. We propose a novel framework centered on dynamic entropy weighting, which introduces two new algorithms: Group Token Policy Optimization (GTPO) for precise, token-level supervision, and a computationally efficient variant, Sequence-Level GRPO (GRPO-S). Our approach repurposes policy entropy as a proxy for model uncertainty to concentrate the learning signal at critical decision points, thereby enabling principled, fine-grained credit assignment. Extensive experiments demonstrate that our methods consistently outperform strong DAPO and GRPO baselines across multiple reasoning benchmarks, confirming the efficacy of the proposed entropy-weighting mechanism. Ultimately, our findings suggest that harnessing and directing model uncertainty is a promising frontier for developing the next generation of powerful and reliable AI systems, as demonstrated qualitatively in the case study in Appendix F.

## Impact Statement

This work introduces a dynamic entropy-weighting framework to address coarse-grained credit assignment in aligning large language models for complex reasoning. By using policy entropy as a signal of model uncertainty, the proposed methods concentrate the learning feedback on critical decision points, which can lead to more data-efficient and targeted optimization. The potential positive societal impact lies in enabling more reliable and sample-efficient alignment of reasoning models. This may contribute to the development of safer and more trustworthy AI assistants in domains where faithful step-by-step reasoning is important, such as education, scientific discovery, and decision support systems. At the same time, careful calibration of entropy-based uncertainty is advisable to avoid inadvertently reinforcing undesirable reasoning patterns.

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

# A. Background and Related Work

## A.1. The Evolution of LLM Alignment Algorithms

LLM alignment techniques aim to make model behavior conform to human expectations and values. The field is initially dominated by PPO-based RLHF. This classic paradigm consists of three stages: supervised fine-tuning (SFT), reward model training, and reinforcement learning optimization. Despite its power, its process is complex, sensitive to hyperparameters, and often unstable during training. To overcome these challenges, the research community shifts towards more direct optimization methods. Direct Preference Optimization (DPO) is a landmark work that cleverly transformed the reward maximization problem into a simple classification loss, completely bypassing explicit reward modeling and the RL process (Rafailov et al., 2023). The success of DPO spawns a series of variants, such as ODPO (Amini et al., 2024), which considers the strength of preferences, and Preference Tuning LLMs with TRL (Hugging Face, 2024), which aims to solve overfitting, collectively advancing the RL-free alignment paradigm.

## A.2. The Rise of Value-Function-Free Policy Optimization

Our work builds directly on value-function-free policy optimization methods. **Group Relative Policy Optimization (GRPO)**, introduced by DeepSeekMath, is a representative of this direction (Shao et al., 2024). The core mechanism of GRPO is: for a given prompt, sample a group of $G$ sequences from the current policy, and then use the average reward within this group as a baseline to calculate the advantage for each sequence (Kilcher, 2024). This design eliminates the need for a separate value function network, greatly reducing memory consumption and computational complexity, which leds to great success in tasks like mathematical reasoning. However, the original GRPO is also sensitive to reward noise and can be unstable during training, which prompts subsequent research for improvements.

## A.3. A Technical Comparison with the DAPO Baseline

**Decoupled Clip and Dynamic sAmpling Policy Optimization (DAPO)** (Yu et al., 2025) is currently the state-of-the-art (SOTA) method for GRPO-style training in the open-source community. DAPO significantly improves the performance and stability of GRPO by introducing four key techniques: 1) **Clip-Higher**: Encourages model exploration and prevents entropy collapse by relaxing the upper bound of the PPO clipping range. 2) **Dynamic Sampling**: Filters out sample groups that are either all successful or all failures, ensuring that each training batch contains effective gradient signals, thus improving training efficiency. 3) **Token-Level Policy Gradient Loss**: A core improvement of DAPO, its objective function averages the loss over all tokens in a batch, rather than first summing within a sequence and then averaging across sequences as in the original GRPO (Yu et al., 2025). 4) **Overlong Reward Penalty**: Penalizes excessively long generated sequences to reduce reward noise.

Our work shares some motivations with DAPO. Specifically, in Appendix B, through variance analysis, it is proven that the token-level loss normalization method used by DAPO (i.e., a single average over all tokens) is statistically superior to GRPO's two-stage averaging method. This provides a theoretical basis for our adoption of a similar loss function structure. However, our core contribution is fundamentally different from DAPO's. DAPO's token-level loss addresses the normalization of the loss *calculation*, but its advantage term $\hat{A}_{i,t}$ remains **constant** for all tokens $t$ within a given sequence $i$. This means DAPO does not solve the fundamental **credit assignment problem** raised in the introduction. Our work, particularly the GTPO algorithm, directly reconstructs the reward signal itself by introducing a dynamic, non-uniform token-level reward $\tilde{r}_{i,t}$, thereby achieving true fine-grained credit assignment. In short, DAPO optimizes *how to sum the losses*, while the content of the loss terms themselves is optimized.

## A.4. Entropy as a Heuristic for Cognitive Effort in LLMs

Using model entropy as a measure of uncertainty has a long history in the machine learning field (Wang et al., 2025c). Recent research shows that during the reasoning process of LLMs, the entropy of the model's generated probability distribution is highly correlated with cognitive uncertainty. For example, Cheng et al. (2025) find that in successful reasoning paths, high-entropy regions often correspond to steps where the model engages in meaningful exploration and critical logical reasoning. This finding provides strong support for our use of entropy as a heuristic for credit assignment and forms the cornerstone of our methodology.

# B. Theoretical Analysis and Proofs

For the purpose of theoretical variance analysis, we assume the reward signals associated with individual tokens are independent and identically distributed (i.i.d.). While token generation is auto-regressive, this simplification allows us to formally derive the variance reduction properties of the global mean estimator.

## B.1. Variance Comparison of Two Mean Calculation Methods

This section provides a detailed proof to show that when estimating the mean of a random variable, directly taking the total mean of all samples (Method 2) is superior to first calculating subgroup means and then averaging them (Method 1).

Let there be a random variable $X$ with mean $\mathbb{E}[X] = \mu$ and variance $Var(X) = \sigma^2$. There are $m$ independent groups, and for the $i$-th group, $n_i$ independent and identically distributed samples are drawn to obtain the sample set $\{x_i^{(1)}, x_i^{(2)}, ..., x_i^{(n_i)}\}$.

**Method 1: First Compute Subgroup Means, Then Average**    For each subgroup $i$ ($1 \leq i \leq m$), its sample mean is computed as follows:

$$\overline{x}_i = \frac{1}{n_i} \sum_{j=1}^{n_i} x_i^{(j)} \tag{15}$$

The expectation of each $\overline{x}_i$ is $\mathbb{E}[\overline{x}_i] = \mu$, and its variance is $Var(\overline{x}_i) = \frac{\sigma^2}{n_i}$. Then, the mean of $X$ is estimated as the average of these subgroup means:

$$\hat{X}_1 = \frac{1}{m} \sum_{i=1}^{m} \overline{x}_i \tag{16}$$

The expectation of $\hat{X}_1$ is $\mathbb{E}[\hat{X}_1] = \frac{1}{m} \sum_{i=1}^{m} \mathbb{E}[\overline{x}_i] = \mu$, which is an unbiased estimator. Its variance is:

$$Var(\hat{X}_1) = \frac{\sigma^2}{m^2} \sum_{i=1}^{m} \frac{1}{n_i} \tag{17}$$

**Method 2: Directly Compute the Grand Mean of All Samples**    The total number of samples is $N = \sum_{i=1}^{m} n_i$. The mean of all samples is computed directly:

$$\hat{X}_2 = \frac{1}{N} \sum_{i=1}^{m} \sum_{j=1}^{n_i} x_i^{(j)} \tag{18}$$

The expectation of $\hat{X}_2$ is $\mathbb{E}[\hat{X}_2] = \mu$, also an unbiased estimator. Its variance is:

$$Var(\hat{X}_2) = \frac{\sigma^2}{N} = \frac{\sigma^2}{\sum_{i=1}^{m} n_i} \tag{19}$$

**Comparing $Var(\hat{X}_1)$ and $Var(\hat{X}_2)$**    According to the Arithmetic Mean-Harmonic Mean (AM-HM) inequality, for any set of positive numbers $n_1, ..., n_m$, the following holds:

$$\frac{\sum_{i=1}^{m} n_i}{m} \geq \frac{m}{\sum_{i=1}^{m} \frac{1}{n_i}} \tag{20}$$

This means the arithmetic mean is greater than or equal to the harmonic mean, with equality holding if and only if all $n_i$ are equal. Since $A \geq H$, it follows that $\frac{1}{A} \leq \frac{1}{H}$. Therefore,

$$Var(\hat{X}_2) \leq Var(\hat{X}_1) \tag{21}$$

It shows that Method 2 is statistically superior because it provides an estimator with smaller (or equal) variance. In the context of RL training for LLMs, $m$ corresponds to the number of sequences in a batch, $G$, and $n_i$ corresponds to the length of the $i$-th sequence, $|o_i|$. Since generated sequence lengths are typically different, $Var(\hat{X}_2) < Var(\hat{X}_1)$.

## B.2. Unifying the objective function of GRPO at Token-Level

However, the previous proof is based on the assumption that the random variables are uniformly distributed. For a more precise proof, a substitution needs to be performed on the random variables.

Continuing from the random variable $X$ above, consider the random variable $Y = f(X)$, where $f$ does not have a specific functional form, but given a value of $X$, a deterministic value of $Y$ (analogous to a neural network) can be obtained . For a specific sample $x_i^{(j)}$, the corresponding value of $Y$ can be obtained as:

$$y_i^{(j)} = f(x_i^{(j)}) = \frac{\pi_\theta(o_{i,j}|q, o_{i,<j})}{\pi_{\theta_{old}}(o_{i,j}|q, o_{i,<j})}(x_i^{(j)} - c),$$

where $c$ is a constant, corresponding to mean($\{R_i\}_{i=1}^G$).

If $\bar{y}_i = \frac{1}{n_i} \sum_{j=1}^{n_i} y_i^{(j)}$ is defined and use the two methods from above to estimate $[Y]$, an identical proof allows us to obtain:

$$Var(\hat{Y}_2) \le Var(\hat{Y}_1).$$

This completes the proof, leading to the following conclusion.

**Conclusion B.1.** *If unifying GRPO at the token-level is considered, a single average $\frac{1}{\sum|o_i|}\sum$ should be used rather than a two-stage average $\frac{1}{G}\sum\frac{1}{|o_i|}\sum$. Therefore, the leading coefficient of GTPO, $\frac{1}{\sum|o_i|}$, is superior to that of GRPO.*

Next, the mean($\{R_i\}_{i=1}^G$) part of the GRPO objective function is analyzed. First, the ideal state of the advantage function is known to be $A = Q - V$, where $Q$ is the action-value and $V$ is the state-value. Therefore, during sampling, Q and V need to be estimated as accurately as possible. The analysis in A.1 for $Y$ is actually an analysis of the estimation method for $Q$. DAPO already modify this, but changing the estimation method for $V$ can be considered. The term mean($\{R_i\}_{i=1}^G$) is the estimate for $V$. Theoretically, there is a more accurate estimation method, which is proven below.

Currently, the way GRPO and DAPO assign rewards to each token in a sequence is by taking the reward from the last token and assigning it to all preceding tokens in that sequence. The final sampling result is equivalent to the result obtained from the following sampling method: $G$ groups of samples are sampled, where each group $o_i$ corresponds to a set of samples $\{r_i^{(1)}, r_i^{(2)}, \ldots, r_i^{(|o_i|)}\}$. The arithmetic mean for each group is then taken to get $\bar{r}_i = \frac{1}{|o_i|}\sum_{j=1}^{|o_i|} r_i^{(j)}$, and then the collected sample for each group is assumed to be $\{\bar{r}_i, \bar{r}_i, \ldots, \bar{r}_i\}$ ($|o_i|$ times).

Without confusion, the notation $\{r_i^{(1)}, \ldots, r_i^{(|o_i|)}\}$ is still used to to represent the set of samples corresponding to $o_i$, but it must be noted that the relation $r_i^{(1)} = r_i^{(2)} = \cdots = r_i^{(|o_i|)} = \bar{r}_i$ holds.

Since the problem is considered at the token-level, the $R$ (reward) in mean($R$) should also be at the token-level, not simply at the sequence-level. The length of the sequence (i.e., the number of tokens) cannot be ignored just because the default sample values within each sequence are identical. The reason is as follows: since the sampling of rewards here is all i.i.d., it is completely equivalent to the analysis of the random variable $X$ in Appendix B.1. Here, i.i.d. samples of $R$ are being dealt with. Let the following be set:

$$\hat{R}_1 := \frac{1}{G}\sum_{i=1}^G \bar{r}_i, \quad \hat{R}_2 := \frac{1}{\sum|o_i|}\sum_{i=1}^G\sum_{j=1}^{|o_i|} r_i^{(j)}.$$

If the reward corresponding to the last token is defined as $r_i$, the following notation consistent with GRPO is used:

$$\hat{R}_1 = \frac{1}{G}\sum_{i=1}^G r_i, \quad \hat{R}_2 = \frac{\sum|o_i|r_i}{\sum|o_i|}.$$

Based on the previous proof for $X$, it can be easily concluded that $Var(\hat{R}_2) \le Var(\hat{R}_1)$. This completes the proof, leading to the following conclusion.

**Conclusion B.2.** *If unifying GRPO at the token-level is considered, mean($\{R_i\}_{i=1}^G$) in GRPO should be replaced with*

$$mean(R) := \frac{\sum_{i=1}^G |o_i|r_i}{\sum_{i=1}^G |o_i|}.$$

## B.3. Theoretical Analysis of Reward Conservation and Convergence

In this section, we provide a formal proof demonstrating that the Dynamic Entropy Weighting mechanism in GTPO preserves the total positive reward mass of the original GRPO objective. This conservation property ensures that the expected policy gradient direction remains consistent with the ground-truth reward maximization, thereby guaranteeing that the positive reward component of GTPO optimizes towards the same global objective as the baseline.

**Proposition B.3** (Conservation of Positive Reward Mass). *Let $O_t^+$ be the set of successful sequences active at timestep $t$ within a sampled group, with cardinality $d_t = |O_t^+|$. Under the GTPO reward shaping mechanism defined in Eq.(3), if the hyperparameters satisfy $\alpha_1 + \alpha_2 = 1$, then the sum of the shaped positive rewards equals the sum of the original binary rewards at every timestep:*

$$\sum_{i \in O_t^+} \tilde{r}_{i,t}^+ = \sum_{i \in O_t^+} r_i. \tag{22}$$

*Proof.* Recall the definition of the shaped token reward for a successful sequence $o_i \in O^+$ at timestep $t$, as provided in the GTPO formulation:

$$\tilde{r}_{i,t}^+ = \alpha_1 r_i + \alpha_2 \frac{H_{i,t}}{\sum_{k \in O_t^+} H_{k,t}} \cdot d_t, \tag{23}$$

where $r_i = 1$ for all successful sequences, $H_{i,t}$ is the policy entropy, and the summation in the denominator is taken over all active successful sequences at timestep $t$.

We compute the sum of shaped rewards over all sequences $i \in O_t^+$:

$$\sum_{i \in O_t^+} \tilde{r}_{i,t}^+ = \sum_{i \in O_t^+} \left( \alpha_1 \cdot 1 + \alpha_2 \frac{H_{i,t}}{\sum_{k \in O_t^+} H_{k,t}} \cdot d_t \right) \tag{24}$$

$$= \sum_{i \in O_t^+} \alpha_1 + \sum_{i \in O_t^+} \left( \alpha_2 d_t \frac{H_{i,t}}{\sum_{k \in O_t^+} H_{k,t}} \right). \tag{25}$$

The first term simplifies directly since there are $d_t$ terms:

$$\sum_{i \in O_t^+} \alpha_1 = \alpha_1 d_t. \tag{26}$$

For the second term, we factor out the constants $\alpha_2$ and $d_t$, observing that the sum of the normalized entropies is unity:

$$\sum_{i \in O_t^+} \left( \alpha_2 d_t \frac{H_{i,t}}{\sum_{k \in O_t^+} H_{k,t}} \right) = \alpha_2 d_t \frac{\sum_{i \in O_t^+} H_{i,t}}{\sum_{k \in O_t^+} H_{k,t}} = \alpha_2 d_t \cdot 1 = \alpha_2 d_t. \tag{27}$$

Combining these results:

$$\sum_{i \in O_t^+} \tilde{r}_{i,t}^+ = \alpha_1 d_t + \alpha_2 d_t = (\alpha_1 + \alpha_2) d_t. \tag{28}$$

Given the constraint $\alpha_1 + \alpha_2 = 1$, we arrive at:

$$\sum_{i \in O_t^+} \tilde{r}_{i,t}^+ = d_t = \sum_{i \in O_t^+} r_i. \tag{29}$$

Since the original reward $r_i$ is uniformly 1 for all successful sequences, its sum is exactly the count $d_t$. This proves that the total positive reward mass distributed at each timestep is strictly conserved.

$\square$

**Proposition B.4** (Asymptotic Consistency). *Let $\Delta\mathcal{J}(\theta) = \nabla\mathcal{J}_{GTPO}^+(\theta) - \nabla\mathcal{J}_{GRPO}(\theta)$ be the gradient bias introduced by the entropy-weighted reward shaping, where $\mathcal{J}_{GTPO}^+(\theta)$ is the positive reward component of GTPO. Assume the policy $\pi_\theta$ satisfies the Entropy Consolidation Condition: as training iteration $k \to \infty$, the variation in token entropy among successful sequences diminishes, i.e., for any two successful sequences $o_i, o_j \in O^+$, $\lim_{k\to\infty} \frac{H_{i,t}}{H_{j,t}} = 1$ (almost surely, assuming regularization $\epsilon > 0$ prevents singularity). Then, the gradient bias vanishes asymptotically:*

$$\lim_{k\to\infty} \|\Delta\mathcal{J}(\theta_k)\| = 0. \tag{30}$$

*Proof.* Recall that the GTPO reward $\tilde{r}_{i,t}$ is defined as:

$$\tilde{r}_{i,t} = \alpha_1 r_i + \alpha_2 \frac{H_{i,t}}{\sum_{p\in O_t^+} H_{p,t}} d_t. \tag{31}$$

Since $r_i = 1$ for all successful sequences, the original GRPO reward is constant 1. The deviation term (exploration bias) for a specific sequence $o_i$ is:

$$\delta_{i,t} = \tilde{r}_{i,t} - r_i = \alpha_1 + \alpha_2 d_t \frac{H_{i,t}}{\sum_{p=1}^{d_t} H_{p,t}} - 1. \tag{32}$$

Substituting $\alpha_1 = 1 - \alpha_2$:

$$\delta_{i,t} = (1 - \alpha_2) + \alpha_2 d_t \frac{H_{i,t}}{\sum_p H_{p,t}} - 1 = \alpha_2 \left( \frac{d_t H_{i,t}}{\sum_p H_{p,t}} - 1 \right). \tag{33}$$

Let $\bar{H}_t = \frac{1}{d_t} \sum_{p\in O_t^+} H_{p,t}$ be the arithmetic mean of entropies for successful sequences at step $t$. The term can be rewritten as:

$$\delta_{i,t} = \alpha_2 \left( \frac{H_{i,t}}{\bar{H}_t} - 1 \right). \tag{34}$$

Under the *Entropy Consolidation* assumption, as $k \to \infty$, the entropies of successful paths converge to a uniform level (typically approaching 0 due to policy collapse, or stabilizing at a intrinsic uncertainty level). In either case, the ratio of individual entropy to the group mean approaches unity:

$$\lim_{k\to\infty} \frac{H_{i,t}}{\bar{H}_t} = 1. \tag{35}$$

Consequently, the reward deviation vanishes:

$$\lim_{k\to\infty} \delta_{i,t} = \alpha_2(1 - 1) = 0. \tag{36}$$

The gradient difference is the expectation of this deviation weighted by the score function:

$$\Delta\mathcal{J}(\theta) = \mathbb{E}_{\tau\sim\pi_\theta} \left[ \sum_t \delta_{i,t} \nabla \log \pi_\theta(o_{i,t}|\cdot) \right]. \tag{37}$$

Since $\delta_{i,t} \to 0$ uniformly for all successful sequences, by the linearity of expectation (and assuming bounded gradients), we conclude:

$$\lim_{k\to\infty} \Delta\mathcal{J}(\theta_k) = 0. \tag{38}$$

This confirms that while the positive reward component of GTPO encourages exploration during the transient phase where entropy variance is high, it asymptotically reduces to the standard GRPO objective, ensuring valid convergence.

$\square$

## B.4. Analysis of GRPO-S

Analogous to the theoretical analysis of GTPO in Appendix B.3, we now present the analysis for GRPO-S. Similar to the token-level analysis, we establish that the sequence-level reward shaping in GRPO-S preserves the fundamental properties of the baseline estimator: reward mass conservation and asymptotic gradient consistency.

**Proposition B.5** (Sequence-Level Reward Conservation). *Let $O^+ = \{o_1, \ldots, o_n\}$ be the set of $n$ successful sequences in a sampled group. Let $\hat{H}_i$ denote the average entropy of sequence $o_i$. Under the GRPO-S reward shaping defined in Eq. (9), if $\beta_1 + \beta_2 = 1$, then the sum of shaped positive sequence rewards equals the sum of original rewards:*

$$\sum_{i=1}^{n} \hat{r}_i^+ = \sum_{i=1}^{n} r_i = n. \tag{39}$$

*Proof.* The GRPO-S shaped reward for a successful sequence $o_i$ is given by:

$$\hat{r}_i^+ = \beta_1 r_i + \beta_2 \frac{\hat{H}_i}{\sum_{k=1}^{n} \hat{H}_k} \cdot n, \tag{40}$$

where $r_i = 1$ for all $o_i \in O^+$. Summing over all $n$ successful sequences:

$$\sum_{i=1}^{n} \hat{r}_i^+ = \sum_{i=1}^{n} \left( \beta_1 \cdot 1 + \beta_2 n \frac{\hat{H}_i}{\sum_{k=1}^{n} \hat{H}_k} \right) \tag{41}$$

$$= n\beta_1 + \beta_2 n \sum_{i=1}^{n} \frac{\hat{H}_i}{\sum_{k=1}^{n} \hat{H}_k}. \tag{42}$$

Observing that the summation term $\sum_{i=1}^{n} \frac{\hat{H}_i}{\sum_{k=1}^{n} \hat{H}_k} = \frac{\sum \hat{H}_i}{\sum \hat{H}_k} = 1$, we have:

$$\sum_{i=1}^{n} \hat{r}_i^+ = n\beta_1 + n\beta_2 = n(\beta_1 + \beta_2). \tag{43}$$

With the constraint $\beta_1 + \beta_2 = 1$, it follows that $\sum \hat{r}_i^+ = n$, which matches the sum of the ground-truth rewards $\sum r_i$. Thus, the total positive reward signal used for the advantage calculation is conserved at the group level.

$\square$

**Proposition B.6** (Asymptotic Gradient Consistency of GRPO-S). *Let $\mathcal{J}_{GRPO}(\theta)$ and $\mathcal{J}_{GRPO\text{-}S}(\theta)$ be the expected objective functions under the original and shaped rewards, respectively. Assume the following regularity conditions:*

1. ***Boundedness:*** *The policy score function $\|\nabla_\theta \log \pi_\theta(o)\|$ is bounded by a constant $M$ for all feasible $o$.*

2. ***Entropy Consolidation:*** *As training progresses ($k \to \infty$), the relative entropy variance among successful sequences vanishes: $\sup_{i \in O^+} |\frac{\hat{H}_i}{\bar{H}} - 1| \xrightarrow{P} 0$.*

*Then, the gradient of the GRPO-S objective converges to the GRPO gradient in $L_1$ norm:*

$$\lim_{k \to \infty} \|\nabla \mathcal{J}_{GRPO\text{-}S}^+(\theta_k) - \nabla \mathcal{J}_{GRPO}(\theta_k)\|_{L_1} = 0. \tag{44}$$

*Proof.* Let $O = \{o_1, \ldots, o_G\}$ be the sampled group. The gradient estimator for GRPO is:

$$g(\theta) = \frac{1}{G} \sum_{i=1}^{G} A_i \nabla_\theta \log \pi_\theta(o_i), \quad \text{where } A_i = \frac{r_i - \mu}{\sigma}. \tag{45}$$

Similarly, the gradient estimator for GRPO-S is:

$$\hat{g}(\theta) = \frac{1}{G} \sum_{i=1}^{G} \hat{A}_i \nabla_\theta \log \pi_\theta(o_i), \quad \text{where } \hat{A}_i = \frac{\hat{r}_i - \hat{\mu}}{\hat{\sigma}}. \tag{46}$$

Here, $\mu, \sigma$ are the mean and standard deviation of the original rewards $\{r\}$, and $\hat{\mu}, \hat{\sigma}$ are of the shaped rewards $\{\hat{r}\}$.

**Step 1: Convergence of the Reward Signal.** Recall from Proposition B.5 that the shaped reward is $\hat{r}_i = \beta_1 r_i + \beta_2 \frac{\hat{H}_i}{\bar{H}} r_i$ (since $r_i = 0$ for incorrect answers, this form holds generally if we define deviation on the correct subset). The deviation is $\delta_i = \hat{r}_i - r_i = \beta_2 r_i (\frac{\hat{H}_i}{\bar{H}} - 1)$. Under the *Entropy Consolidation* assumption, for any $\epsilon > 0$, there exists a step $K$ such that for all $k > K$, $|\frac{\hat{H}_i}{\bar{H}} - 1| < \epsilon$. Thus, the shaped rewards converge pointwise to the original rewards:

$$\lim_{k \to \infty} |\hat{r}_i - r_i| = 0. \tag{47}$$

**Step 2: Continuity of the Advantage Function.** The Advantage function $A(r) = \frac{r - \text{mean}(r)}{\text{std}(r)}$ is a continuous mapping with respect to the input vector $\mathbf{r} \in \mathbb{R}^G$, provided $\text{std}(r) > 0$ (which is true in practice as we sample diverse groups or use $\epsilon$-regularization). Since $\hat{r}_i \to r_i$ for all $i$, by the Continuous Mapping Theorem:

$$\lim_{k \to \infty} |\hat{\mu} - \mu| = 0 \quad \text{and} \quad \lim_{k \to \infty} |\hat{\sigma} - \sigma| = 0. \tag{48}$$

Consequently, the shaped advantage converges to the original advantage:

$$\lim_{k \to \infty} |\hat{A}_i - A_i| = 0. \tag{49}$$

**Step 3: Convergence of the Expected Gradient.** Consider the norm of the difference between the expected gradients:

$$\|\Delta \mathcal{J}\|_{L_1} := \|\nabla \mathcal{J}_{\text{GRPO-S}}^+(\theta_k) - \nabla \mathcal{J}_{\text{GRPO}}(\theta_k)\|_{L_1} \tag{50}$$

$$= \left\| \mathbb{E}_{O \sim \pi} \left[ \frac{1}{G} \sum_{i=1}^{G} (\hat{A}_i - A_i) \nabla_\theta \log \pi_\theta(o_i) \right] \right\| \tag{51}$$

$$\leq \mathbb{E}_{O \sim \pi} \left[ \frac{1}{G} \sum_{i=1}^{G} |\hat{A}_i - A_i| \cdot \|\nabla_\theta \log \pi_\theta(o_i)\| \right]. \tag{52}$$

By the *Boundedness* assumption, $\|\nabla \log \pi\| \leq M$. Thus:

$$\|\Delta \mathcal{J}\|_{L_1} \leq M \cdot \mathbb{E}_{O \sim \pi} \left[ \frac{1}{G} \sum_{i=1}^{G} |\hat{A}_i - A_i| \right]. \tag{53}$$

Let $X_k = \frac{1}{G} \sum |\hat{A}_i - A_i|$ be the random variable representing the average advantage deviation at step $k$. We established in Step 2 that $X_k \xrightarrow{a.s.} 0$. Since the rewards and advantages are bounded (clipped in implementation), $X_k$ is dominated by a bounded constant. By the **Lebesgue Dominated Convergence Theorem**, we can interchange the limit and the expectation:

$$\lim_{k \to \infty} \|\Delta \mathcal{J}\|_{L_1} \leq M \cdot \mathbb{E} \left[ \lim_{k \to \infty} X_k \right] = M \cdot \mathbb{E}[0] = 0. \tag{54}$$

This proves that the gradient of the positive reward component of GRPO-S is an asymptotically consistent estimator of the GRPO gradient.

$\square$

Based on Propositions B.5 & B.6, and following a proof analogous to that of Theorem 2.4, we derive the following theorem:

**Theorem B.7** (The Same Global Optimum - GRPO-S). *GRPO-S shares the same global optimum as DAPO.*

*Remark* B.8. Note that while Proposition B.5 assumes $\beta_1 + \beta_2 = 1$, this condition is not strictly necessary. The proofs remain valid even if $\beta_1 + \beta_2 \neq 1$, as the term $\beta_1 + \beta_2$ merely acts as a constant scaling factor that does not affect the validity of the derivation. In our experimental setup, the sum $\beta_1 + \beta_2$ is configured to be close to 1, but strict equality is not enforced.

## C. Alternative Definitions via Geometric Mean

Recall that we used Eq.(11) to define the sequence-level importance weight $\hat{w}_i(\theta)$, which is an arithmetic mean. In fact, we can define the sequence-level importance weight via geometric mean: $\hat{w}_i(\theta) = (\prod_{t=1}^{|o_i|} w_{i,t}(\theta))^{1/|o_i|}$. We can employ the geometric mean as a low-variance proxy for the sequence-level importance weight. Although this introduces bias, it acts as a geometric baseline that stabilizes training for long sequences. Similarly, we have alternative definitions for other key concepts via geometric mean:

$$\tilde{r}_{i,t}^+ = \alpha_1 r_i + \alpha_2 \frac{H_{i,t}}{(\prod_{k=1}^n H_{k,t})^{1/d_t}}, \qquad \tilde{r}_{j,t}^- = \alpha_1(-1) + \alpha_2 \frac{1/H_{j,t}}{(\prod_{k=1}^m 1/H_{k,t})^{1/h_t}}(-1),$$

$$\hat{r}_i^+ = \beta_1 r_i + \beta_2 \frac{\hat{H}_i}{(\prod_{k=1}^n \hat{H}_k)^{1/n}}, \qquad \hat{r}_j^- = \beta_1(-1) + \beta_2 \frac{1/\hat{H}_j}{(\prod_{k=1}^m 1/\hat{H}_k)^{1/m}}(-1),$$

$$\hat{H}_k = (\prod_{t=1}^{|o_k|} H_{k,t})^{1/|o_k|}, \qquad \hat{w}_i(\theta) = (\prod_{t=1}^{|o_i|} w_{i,t}(\theta))^{1/|o_i|}.$$

Crucially, we note a specific distinction here: if a sequence $o_*$ has length less than $t$, its $H_{*,t}$ or $1/H_{*,t}$ is treated as 1, instead of 0, to ensure the denominator remains non-zero.

## D. Group Relative Overlong Punishment: A Heuristic for Length Control

To solve the problem of declining accuracy on simple tasks and consistent failures on complex ones, a differential penalty is proposed to be applied to the response length based on the classified difficulty of each task. Details are as follows.

First, for a given question q, a group of responses $\{o_i\}_{i=1}^G$ is sampled. Following the previous notation, this group is assumed to consist of n correct responses and m incorrect responses. **Easy Question** and **Hard Question** can then be distinguished.

**Easy Question:** If $\frac{n}{n+m} \geq \gamma_1$, then $q$ is called an easy question. Where $0 < \gamma_1 < 1$.

**Hard Question:** If $\frac{n}{n+m} \leq \gamma_2$, then $q$ is called a hard question. Where $\gamma_2 = 1 - \gamma_1$.

Then the **Group Relative Overlong Punishment** can be defined:

- Let $L_1^+ = \min\{|o_i|, 1 \leq i \leq n\}$, and let $L_2^+ = \max\{|o_i|, 1 \leq i \leq n\}$. Then we define

$$L^+ = \max\{\frac{L_1^+ + L_2^+}{2}, \bar{L}^+\},$$

  where $\bar{L}^+ = \frac{\sum_{i=1}^n |o_i|}{n}$. Then for $q$ is an **easy question**, a **Group Relative Overlong Punishment** is set for the **correct responses** as following:

$$R^+(i) = \begin{cases} -\frac{1}{2}\frac{|o_i|-L^+}{L^+} & \text{if } L^+ \leq |o_i| < 2L^+, \\ -\frac{1}{2} & \text{if } 2L^+ \leq |o_i|. \end{cases}$$

- For **hard questions**, no response length punishment is set, or the same punishment as DAPO is set. This is because the goal is to preserve the policy model's ability to produce correct answers to the greatest extent possible.

- Let $L_1^- = \min\{|o_i|, 1 \leq i \leq G\}$, and let $L_2^- = \max\{|o_i|, 1 \leq i \leq G\}$. Then define

$$L^- = \max\{\frac{L_1^- + L_2^-}{2}, \bar{L}^-\},$$

  where $\bar{L}^- = \frac{\sum_{i=1}^G |o_i|}{G}$. For $q$ is a question that is neither easy nor hard, if $n > m$, a **Group Relative Overlong Punishment** is set for the **correct responses** as follows:

$$R^-(i) = \begin{cases} -\frac{1}{2}\frac{|o_i|-L^-}{L^-} & \text{if } L^- \leq |o_i| < 2L^-, \\ -\frac{1}{2} & \text{if } 2L^- \leq |o_i|. \end{cases}$$

If $n \leq m$, a **Group Relative Overlong Punishment** is set for the **incorrect responses** as following:

$$R^-(j) = \begin{cases} -\frac{1}{2}\frac{|o_j|-L^-}{L^-} & \text{if } L^- \leq |o_j| < 2L^-, \\ -\frac{1}{2} & \text{if } 2L^- \leq |o_j|. \end{cases}$$

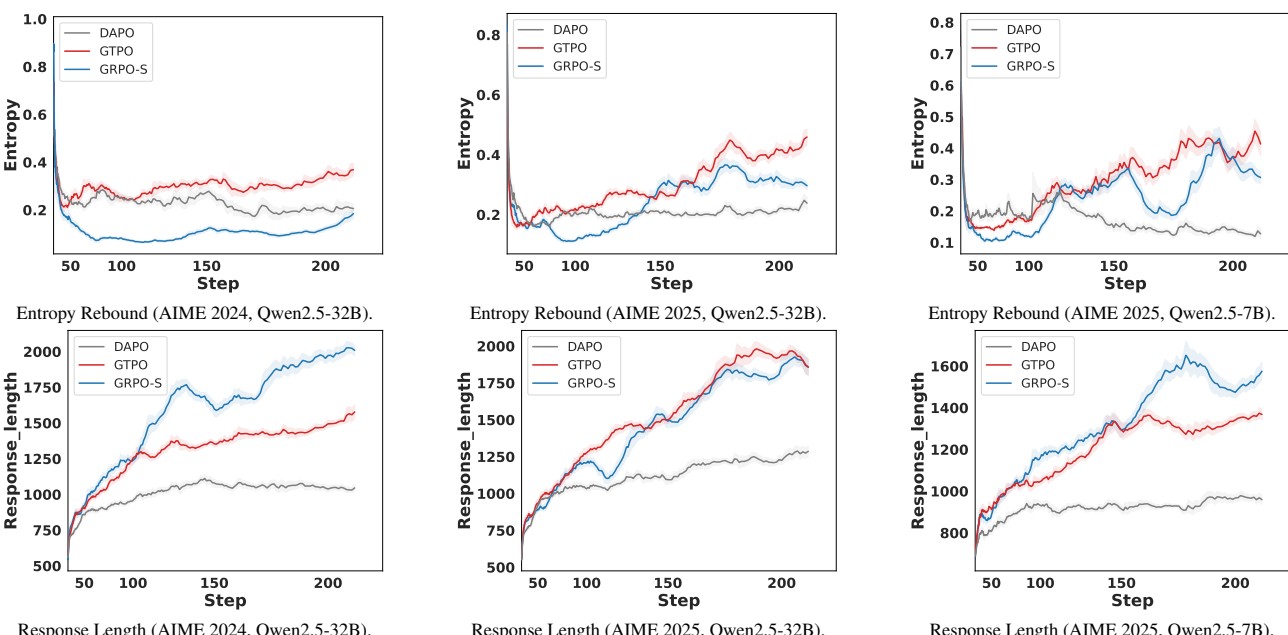

Entropy Rebound (AIME 2024, Qwen2.5-32B).   Entropy Rebound (AIME 2025, Qwen2.5-32B).   Entropy Rebound (AIME 2025, Qwen2.5-7B).

Response Length (AIME 2024, Qwen2.5-32B).   Response Length (AIME 2025, Qwen2.5-32B).   Response Length (AIME 2025, Qwen2.5-7B).

**Figure 6.** The Entropy Rebound Phenomenon and its Effect on Response Length. **Top Row:** The policy entropy trajectories for experiments on (left to right) AIME 2024 with Qwen2.5-32B, AIME 2025 with Qwen2.5-32B, and AIME 2025 with Qwen2.5-7B. Our methods (GTPO, GRPO-S) exhibit a distinct entropy rebound after an initial dip, successfully counteracting the policy collapse observed in the DAPO baseline. **Bottom Row:** The corresponding average response length trajectories. The sustained exploration enabled by the entropy rebound directly manifests as an increase in the average response length, indicating more thorough and diverse reasoning.

# E. Additional Experimental Results and Analysis

## E.1. Analysis of Training Dynamics: Entropy Rebound and Exploration

Figure 6 provides empirical evidence for the core mechanism of our proposed methods. The top row illustrates the "entropy rebound" phenomenon. While all methods initially exhibit a decrease in policy entropy as they learn to exploit correct strategies, the DAPO baseline's entropy continues to decline, indicating convergence to a narrow, deterministic policy, often termed policy collapse. In contrast, both GTPO and GRPO-S show a distinct rebound in entropy. This is direct evidence that our entropy-weighted reward shaping successfully incentivizes the model to maintain exploration. By rewarding uncertainty on successful paths, our methods encourage the model to escape local optima and explore a more diverse set of reasoning pathways. This sustained exploration, as shown in the bottom row, directly results in longer, more detailed responses as the model attempts more thorough lines of reasoning. This establishes a clear causal link from our reward mechanism to superior problem-solving capabilities, as validated by the results in Table 1.

## E.2. Analysis of Generation Characteristics: Response Length and Clipping

Figure 7 displays the response length clip ratio, which is the fraction of generated sequences that reach the maximum token limit. The significantly higher clip ratio for GTPO and GRPO-S serves as further evidence of enhanced exploration. This indicates that our methods encourage the model to generate more elaborate and detailed reasoning chains, often exhausting the available generation budget. This contrasts with the DAPO baseline, where premature policy convergence leads to shorter, less exploratory responses.

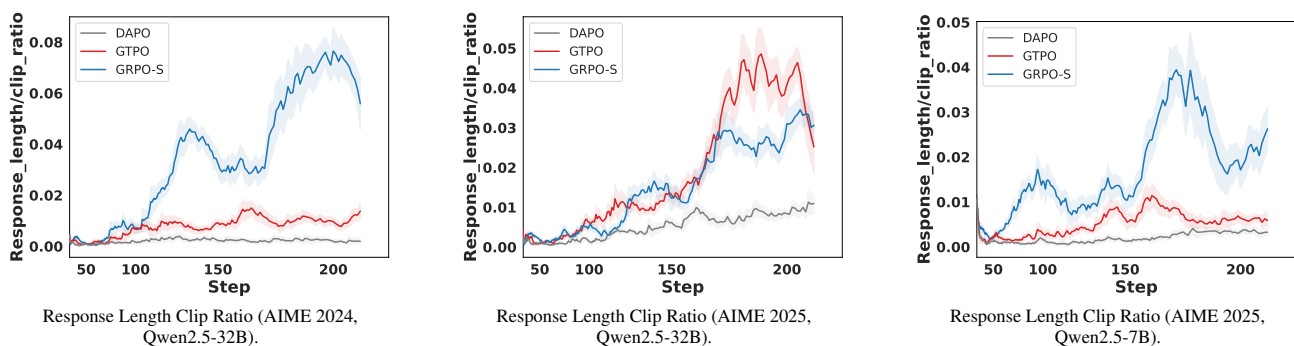

**Figure 7.** Response Length Clip Ratio Trajectories. The plots show the fraction of generated sequences that reached the maximum length limit of 4096 tokens for experiments on (left to right) AIME 2024 with Qwen2.5-32B, AIME 2025 with Qwen2.5-32B, and AIME 2025 with Qwen2.5-7B. The consistently higher clip ratio for GTPO and GRPO-S (around 10%) compared to DAPO provides further evidence of enhanced exploration, as our methods encourage the model to generate more thorough responses that often utilize the full generation budget.

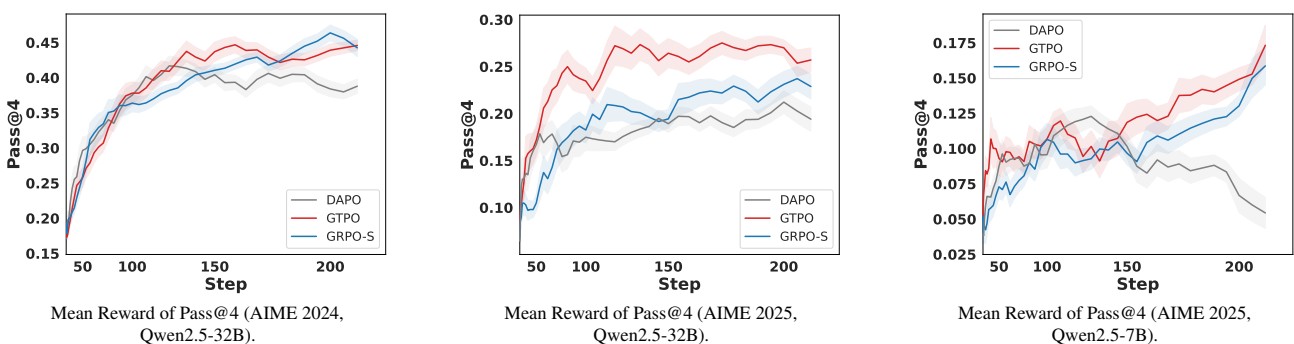

**Figure 8.** Mean Reward Trajectories for Pass@4 on Test Sets. The plots show the mean reward of the top 4 generations for experiments on (left to right) AIME 2024 with Qwen2.5-32B, AIME 2025 with Qwen2.5-32B, and AIME 2025 with Qwen2.5-7B. The trends are consistent with those reported in the main paper, with GTPO and GRPO-S achieving a higher reward ceiling than the DAPO baseline.

### E.3. Complete Reward Trajectories on Test Sets

Figures 8 and 9 present the mean reward trajectories for Pass@4 and Pass@16, respectively. These plots complete the picture presented in Figure 3 of the main paper. The observed trends are highly consistent across all Pass@k metrics: GTPO and GRPO-S consistently achieve a higher final reward ceiling than the DAPO baseline, demonstrating the robustness of our performance gains.

### E.4. Reward Trajectories on Training Sets

Figure 10 shows the mean reward trajectories on the training sets. These curves are crucial for evaluating sample efficiency. The plots indicate that all models, including our proposed methods and the baseline, largely converge within 210 training steps. This demonstrates that the substantial performance improvements achieved by GTPO and GRPO-S are not the result of longer training but are due to a more effective and efficient learning signal derived from our entropy-weighting mechanism.

## F. Qualitative Case Study

We present a qualitative case study to demonstrate the superiority of our GTPO and GRPO-S algorithms, formulated under the **Dynamic Entropy Weighting** framework, over DAPO. Below, we illustrate the actual problem-solving trajectories of these three algorithms on a mathematical problem requiring long-chain reasoning.

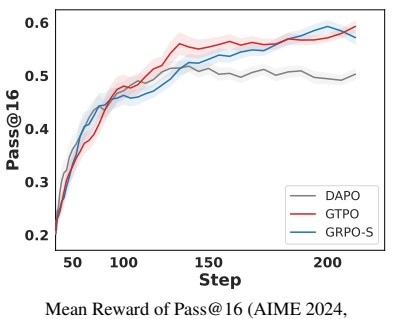 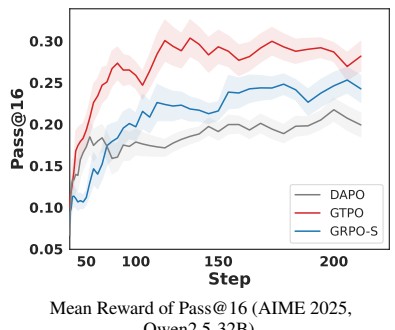 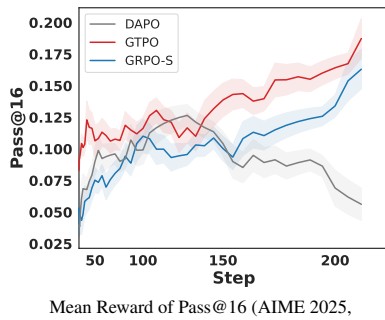

Mean Reward of Pass@16 (AIME 2024, Qwen2.5-32B).  Mean Reward of Pass@16 (AIME 2025, Qwen2.5-32B).  Mean Reward of Pass@16 (AIME 2025, Qwen2.5-7B).

**Figure 9.** Mean Reward Trajectories for Pass@16 on Test Sets. The plots show the mean reward of the top 16 generations for experiments on (left to right) AIME 2024 with Qwen2.5-32B, AIME 2025 with Qwen2.5-32B, and AIME 2025 with Qwen2.5-7B. These results further demonstrate the robust and consistent performance improvements of our methods across different evaluation metrics.

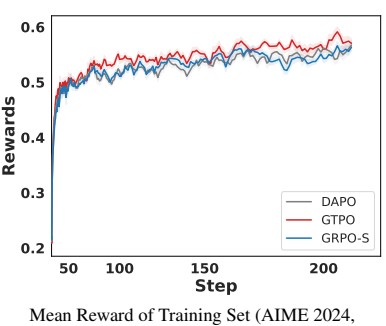 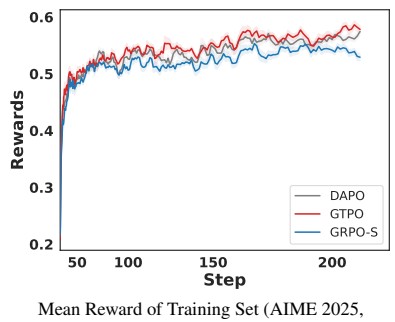 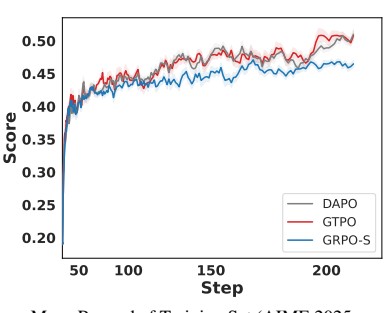

Mean Reward of Training Set (AIME 2024, Qwen2.5-32B).  Mean Reward of Training Set (AIME 2025, Qwen2.5-32B).  Mean Reward of Training Set (AIME 2025, Qwen2.5-7B).

**Figure 10.** Mean Reward Trajectories on Training Sets. The plots show the mean reward on the training data for experiments on (left to right) AIME 2024 with Qwen2.5-32B, AIME 2025 with Qwen2.5-32B, and AIME 2025 with Qwen2.5-7B. The reward curves demonstrate that all models approach convergence on the training data by approximately 210 steps, supporting the claim that the performance benefits of our methods do not come at the cost of reduced sample efficiency.

---

**Problem.**

Alice chooses a set $A$ of positive integers. Then Bob lists all finite nonempty sets $B$ of positive integers with the property that the maximum element of $B$ belongs to $A$. Bob's list has 2024 sets. Find the sum of the elements of $A$.

---

**Solution of GTPO:**

For a given set $A = \{a_1, a_2, \ldots, a_n\}$, where $a_1 < a_2 < \cdots < a_n$, and for any subset $B \subset A$, the maximum element of $B$ must belong to $A$. For a fixed maximum element $a_i \in B \subset A$, the set $B$ can include any subset of the elements $\{1, 2, \ldots, a_i - 1\}$ along with $a_i$. Thus, for a given $a_i$, the number of possible sets $B$ with maximum element $a_i$ is the number of subsets of $\{1, 2, \ldots, a_i - 1\}$ plus the element $a_i$ itself. This is $2^{a_i - 1}$. Therefore, the total number of such sets $B$ is the sum of $2^{a_i - 1}$ for all $a_i \in A$, giving $\sum_{i=1}^{n} 2^{a_i - 1} = 2024$. The task is then to find a set $A = \{a_1, a_2, \ldots, a_n\}$ that satisfies the above equation. To find such $A$, we can express 2024 as a sum of powers of 2.

**Then 2024 is converted to its unique binary representation:**

$$\mathbf{2024_{10} = 11111101000_2}$$
$$= \mathbf{2^{10} + 2^9 + \cdots + 2^6 + 2^5 + 2^3}.$$

Since the binary representation of any integer is unique, we have

$$n = 7, a_7 - 1 = 10, a_6 - 1 = 9, a_5 - 1 = 8, a_4 - 1 = 7, a_3 - 1 = 6, a_2 - 1 = 5, a_1 - 1 = 3.$$

Then $\cdots$, we get $\sum_{i=1}^{n} a_i = 4 + 6 + 7 + 8 + 9 + 10 + 11 = 55$.

Hence, we get the sum of the elements of $A$ is $\boxed{55}$. ✓

**Solution of GRPO-S:**

**Step 1: Understanding the Sets $B$.** Denote the elements of $A$ as $\{a_1, a_2, \ldots, a_n\}$, where $a_1 < a_2 < \cdots < a_n$. According to the problem statement, we know that the maximum element of $B$ must belong to $A$.

**Step 2: Counting the Sets $B$.** For each $a_i \in A$, consider the sets $B$ where $a_i$ is the maximum element. Note that $a_i$ is the maximum element of $B$, then the other elements of $B$ can be any subset of $\{1, 2, \ldots, a_i - 1\}$. The number of subsets of $\{1, 2, \ldots, a_i - 1\}$ is $2^{a_i - 1}$. Since $a_i$ is the maximum element of $B$, $a_i \in B$, which means $B$ is nonempty. Thus, there are $2^{a_i - 1}$ possible sets $B$ for each $a_i$.

**Step 3: Total Number of Sets $B$.** Since the sets $B$ are disjoint based on their maximum elements, the total number of sets $B$ is the sum of the number of sets for each $a_i$. This gives $\sum_{i=1}^{n} 2^{a_i - 1} = 2024$.

**Step 4: Finding the Set $A$.** Then the task is to find a set $A = \{a_1, a_2, \ldots, a_n\}$ such that the sum of $2^{a_i - 1}$ for all $i$ equals 2024. This is equivalent to finding a set of exponents such that the sum of the corresponding powers of 2 equals 2024.

**Let's express 2024 in binary uniquely to identify the exponents:**

$$2024_{10} = 11111101000_2.$$

The unique binary representation tells us that:

$$n = 7, a_7 - 1 = 10, a_6 - 1 = 9, a_5 - 1 = 8, a_4 - 1 = 7, a_3 - 1 = 6, a_2 - 1 = 5, a_1 - 1 = 3.$$

Then $\cdots$, which get $\sum_{i=1}^{n} a_i = 4 + 6 + 7 + 8 + 9 + 10 + 11 = 55$.

Hence, the sum of the elements of $A$ is $\boxed{55}$. ✓

---

**Solution of DAPO:**

Denote $A$ as $\{a_1, a_2, \ldots, a_n\}$, where $a_1 < a_2 < \cdots < a_n$. For each $a_i$, the sets $B$ where the maximum element is $a_i$ can be formed by taking any subset of $\{1, 2, \ldots, a_i - 1\}$ and adding $a_i$ to it. The number of such sets is $2^{a_i - 1}$ because there are $2^{a_i - 1}$ subsets of $\{1, 2, \ldots, a_i - 1\}$. Thus, the total number of sets $B$ is the sum of $2^{a_i - 1}$ for all $a_i \in A$:

$$2024 = 2^{a_1 - 1} + 2^{a_2 - 1} + \cdots + 2^{a_n - 1}$$
$$= 2^{10} + 2^9 + \cdots + 2^3 + 2^2 + 2^0$$

This means $A$ can be chosen such that $a_i - 1$ corresponds to the exponents in the right-hand side of the above equation. Thus, it follows that:

$$a_1 = 1, \ a_2 = 3, \ a_3 = 4, \ a_4 = 5, a_5 = 6, \ a_6 = 7, \ a_7 = 8, \ a_8 = 9, a_9 = 10, \ a_{10} = 11.$$

Therefore, $\sum_{i=1}^{n} a_i = 1 + 3 + 4 + 5 + 6 + 7 + 8 + 9 + 10 + 11 = 64$.

Hence, the sum of the elements of $A$ is $\boxed{64}$. ✗ $>>> 55$

---

**Analysis** This case study provides a compelling qualitative illustration of the practical difference between coarse-grained and fine-grained credit assignment. The DAPO model's failure is characteristic of a system that learns the general structure of a solution (the "template") but lacks logical rigor in its execution. Its coarse-grained reward signal (a single +1 or -1 for the entire sequence) is insufficient to penalize subtle but critical errors like the incorrect binary decomposition. The model can thus become overconfident in a flawed reasoning path. In contrast, the success of the GTPO and GRPO-S models highlights the benefit of an entropy-aware reward signal. Our methods are designed to penalize low-entropy (high-confidence) mistakes, which would directly discourage the kind of confident but incorrect decomposition made by the DAPO model. Simultaneously, by rewarding exploration in successful paths, our methods encourage a more careful and deliberate reasoning process, leading to the discovery of the correct, logically sound solution. This case demonstrates that our framework is key to moving LLMs beyond mere pattern imitation towards robust, verifiable reasoning.

