# OpenReview forum: "GTPO and GRPO-S: Token and Sequence-Level Reward Shaping with Policy Entropy"
_ICML.cc/2026/Conference — ICML 2026 regular_

### Official Review · Reviewer_NKFw · 2026-02-15

**Soundness:** 2
**Presentation:** 3
**Significance:** 3
**Originality:** 2
**Overall Recommendation:** 4
**Confidence:** 3

**Summary:**

This paper introdcues 2 novel algorithms, GTPO and GRPO-S trying to resolve the issue of corase-grained credit assignment in algorithms like GRPO and DAPO. GTPO assigns an entropy-weighted reward to each token and integrates with token-specific advantages, while GRPO-S extends GTPO to sequence level to improve CoT performances and stability. The paper uses rigid theory analysis and some experiments to demonstrate potential advantages of these methods over baselines such as DAPO and GRPO.

**Compliance With Llm Reviewing Policy:**

Affirmed.

**Final Justification:**

The rebuttal addresses most of my concerns, including the actual performance in compared with GSPO. However, I'm still concerned about the originality in the presence of GSPO, since the proposed algorithms are similar to GSPO.

**Key Questions For Authors:**

1. Theorem 2.4 claims that GTPO and DAPO share a global optimum, but the proof only demonstrates the conservation of quality for the positive reward component. For the negative reward component, the penalty mechanism (inverse entropy weighting) alters the relative gradient of the error sequence. Can you explain it?

2. High entropy may indicate more exploration, but it may also indicate unstable training. Can you explain the balance between the two sides of the coin?

3. Have you tried more experiments on different datasets or tasks?

**Limitations:**

yes

**Strengths And Weaknesses:**

Strengths:
1. Soundness: This paper provides detailed analysis on the reasonability of dynamic entropy weighting. The authors also conduct experiments to compare the proposed methods with baselines with detailed training dynamics.

2. Presentation: The paper provides a lot of vivid figures to illustrate the reward assignments and signaling process. And the structure of the paper is also clear and concise.

3. Significance and originality: It identifies an important issue in RLHF. And the proposed algorithms may enlighten further improvement of dynamic entropy weighting-based reward design in the RLHF objectives.

Weaknesses:
1. Soundness:
   - The information for the training datasets is missing.
   - This paper does not consider (and cite) GSPO [1] (published on July 2025), which proposed a similar sequence-level optimization algorithm.
   - It seems that the experiments are only conducted on 1 kind of dataset (potentially dataset in mathematics?) More experiments on different tasks are required to validify the efficiency of the proposed algorithms

2. Presentation: One disadvantage is that some characters in the figures are too small to be recognized.

3. Significance and originality: The originality of GRPO-S is doubted with the absence of GSPO [1]. Moreover, the additional computation in dynamic entropy weighting may downgrade the significance of the proposed methods.

[1] Zheng C, Liu S, Li M, et al. Group sequence policy optimization[J]. arXiv preprint arXiv:2507.18071, 2025.

---

> ### Author Rebuttal · Authors · 2026-03-31
>
> We thank the reviewer for carefully reading and providing constructive suggestions. Below we address each weakness/question in turn.
>
> ---
>
> **Weakness #1 – Soundness**
> - **Information for Training Datasets**: We use the same training set as DAPO, i.e., DAPO-MATH-17K (17.4K math reasoning samples), ensuring fair comparison.
>
> - **Insufficient Experimental Analysis**: We thank the reviewer for suggesting GSPO and include comparisons below:
>
> |                        | AIME 2024 | AIME 2025 | MATH 500 |
> |------------------------|--------|--------|----------|
> | Qwen2.5-7b base        | 16.67  | 8.33   | 50.02    |
> | GSPO                   | 33.67  | 16.33  | **80.20** |
> | GTPO                   | **34.33** | **16.67** | 80.16    |
>
> We further evaluate GTPO and GRPO-S on three OOD reasoning benchmarks: MMLU-Pro [1], GPQA-Diamond [2], and ARC-Challenge [3], including GSPO for comparison. Results show that both GTPO and GRPO-S still maintain a clear advantage on these tasks:
>
> |                        | MMLU-Pro [1] | GPQA-Diamond [2] | ARC-Challenge [3] |
> |------------------------|--------------|----------|--------------------|
> | qwen2.5-base           | 35.42        | 40.91    | 71.19              |
> | GSPO                   | 39.28        | 48.83    | 83.65              |
> | GTPO                   | **40.34**    | **50.13**| **84.08**          |
> | GRPO-S                 | 39.46        | 49.38    | 83.19              |
>
> [1] Mmlu-pro: A more robust and challenging multi-task language understanding benchmark
>
> [2] Gpqa: A graduate-level google-proof q&a benchmark
>
> [3] Think you have Solved Question Answering? Try ARC, the AI2 Reasoning Challenge
>
> ---
>
> **Weakness #2 – Figure readability**
> We thank the reviewer for noting this issue. All figures have been updated to improve readability (text, axes, legends) and will be included in the revision.
>
> ---
>
> **Weakness #3 – Originality and cost**
> We have added GSPO to related work and provided comparisons above. Here we further clarify the **conceptual differences** between our methods and GSPO:
>
> - GSPO applies geometric weighting over full-response token importance weights.
> - GRPO-S uses arithmetic averaging with dynamic entropy-based weighting, enabling differentiated sequence-level scoring.
>
> Thus, the approaches are fundamentally different.
>
> **Computational cost:** Our procedure consists of three steps and does not introduce additional computational overhead:
> - Logits computation: the same as other methods;
> - Entropy computation: parallelizable, $O(V)$ per token;
> - Averaging: no additional complexity.
>
> Hence, the asymptotic time complexity remains unchanged.
>
> ---
>
> **Question #1 – Negative part in Theorem 2.4**
> The theoretical guarantees for the negative part are symmetric with those of the positive part.
>
> - **Conservation**: For active failed sequences at step $t$ (with $\alpha_1+\alpha_2=1$), we have $\sum_{j\in O_t^{-}} \tilde{r}_{j,t}^{-} = -|O_t^{-}|$.
> - **Asymptotic consistency**: $\lim_{k\to\infty} (1/H_{p,t})/(1/H_{q,t})=\frac{1}{\lim_{k\to\infty} H_{p,t}/H_{q,t}}=1$, so the pointwise deviation approaches zero.
> - **Shared global optimum**: By the conservation of total reward mass, GTPO performs a zero-sum redistribution of the DAPO objective. The overall gradient decomposes as
>   $\nabla J_{\text{GTPO}} = \nabla J_{\text{DAPO}} + \mathbb{E}\big[\text{Cov}(\Delta_{\text{entropy}}, \nabla\log\pi_\theta)\big].$
>   This covariance term vanishes at optimality (when entropy variation disappears). Hence the global optimum remains mathematically identical.
>
> ---
>
> **Question #2 – Exploration–stability trade-off**
> We thank the reviewer for raising the important point regarding the trade-off between exploration and training stability. We agree that encouraging high entropy without proper control may affect rollout quality.
>
> However, we do not encourage unbounded high entropy. Instead, we leverage higher entropy to enhance exploration in rollouts, while explicitly controlling it via a scaling parameter ($\alpha_2$ and $\beta_2$). This design prevents the model from exploiting high entropy for reward hacking. Throughout our experiments, we observe that undesirable rollout behaviors (e.g., gibberish or repetition) remain within an acceptable range and do not impact training stability. This is also supported by the smooth training curves presented in the paper. We list three related works [1–3] that similarly use high entropy to encourage exploration:
>
> [1] Reasoning with exploration: An entropy perspective
>
> [2] Beyond the 80/20 rule: High-entropy minority tokens drive effective reinforcement learning for llm reasoning
>
> [3] Arbitrary Entropy Policy Optimization Breaks The Exploration Bottleneck of Reinforcement Learning
>
> ---
>
> **Question #3 – Experiments on different datasets or tasks**
> To address generalization, we include three OOD benchmarks (see Weakness #1), showing that reasoning ability learned from math data transfers to other reasoning domains.

---

> > ### Author Rebuttal · Reviewer_NKFw · 2026-04-01
> >
> > Thank you for your detailed rebuttal. I will raised my score to 4 although I'm still concerned about the originality in the presence of GSPO.

---

> > > ### Author Response · Authors · 2026-04-02
> > >
> > > Thank you very much for taking the time to re-engage with our rebuttal and for raising your score. We sincerely appreciate your willingness to re-evaluate the work, and we are grateful that you found our response helpful enough to move your assessment forward.
> > >
> > > ---
> > >
> > > We completely understand your continued concern regarding originality in light of **GSPO**, and we would like to take this opportunity to offer a bit more clarification to hopefully address that more thoroughly.
> > >
> > > We proposed **GTPO** and **GRPO‑S** because we observed that the effectiveness of the advantage function $A_{i,t}$ in the GRPO objective remains at the sequence level, failing to capture token‑level specificity. To address this, we introduced **GTPO**, which explicitly models token‑level specificity.
> > >
> > > Building on GTPO, we symmetrically considered that each component of the GRPO objective could be degenerated back to the sequence level, leading to **GRPO‑S**. However, the core design of GRPO‑S still lies in using entropy to redistribute rewards, making the sequence‑level advantage functions $\hat{A}_i^+$ or $\hat{A}_i^-$ dynamic even at the sequence level.
> > >
> > > The similarity between GRPO‑S and **GSPO** is that both apply weighting to the importance sampling term in GRPO. Specifically:
> > >
> > > - In **GRPO‑S**, the specific way of weighting the importance sampling term follows Eq. (11) in our paper (we would like to display Eq. (11) here, but despite multiple revisions and attempts, it unfortunately cannot be rendered at this location).
> > >
> > > - In **GSPO**, the importance weight is computed as
> > >
> > >   $$s_i(\theta) = \left( \frac{\pi_{\theta}(y_i \mid x)}{\pi_{\theta_{\text{old}}}(y_i \mid x)} \right)^{\frac{1}{|y_i|}}.$$
> > >
> > > Given the fundamental differences in both the **motivation** behind each algorithm and the **specific form of importance weighting**, GRPO‑S and GSPO are inherently distinct.

---

### Official Review · Reviewer_tZj3 · 2026-03-05

**Soundness:** 3
**Presentation:** 3
**Significance:** 3
**Originality:** 3
**Overall Recommendation:** 4
**Confidence:** 4

**Summary:**

The paper proposes two methods for modifying GRPO using the entropy of model responses to provide a more flexible reward. The first method adds a per-token reward in the form of a normalized component of the entropy in the current token for correct model responses and subtracts the normalized inverse entropy for incorrect responses. The second approach uses the average entropy over all positive generations (resulting in a correct response) and the average inverse of all negative generations (resulting in an incorrect response) separately, and applies a constant addition for each sequence in the group of correct responses and in the group of incorrect responses. The authors compare the method with DAPO and provide theoretical justification for the convergence of their method to the same global optimum. The authors evaluate the method on the AIME24, AIME25, and MATH500 benchmarks where it exhibits better performance according to pass@k metrics (k>1).

**Compliance With Llm Reviewing Policy:**

Affirmed.

**Key Questions For Authors:**

1. Entropy in GRPO typically collapses over time. Do you study its dynamics during training, and if so, how does your method affect its behavior?

2. Do you care about the balance between positive and negative responses within single-example generations? Since your method involves grouping positive and negative responses, this seems likely to impact the stability and convergence of the method.

3. Can you justify the choice of inverse entropy for generations with incorrect answers? Why not use negative entropy, which would potentially eliminate numerical instability when dividing by a low entropy value?

**Limitations:**

yes

**Strengths And Weaknesses:**

**Strengths:**
- The authors address the pressing problem of reward shaping for GRPO-like methods, thereby offering an easy way to add a value function analog to GRPO.
- The authors propose an interesting use of model entropy as a confidence measure for per-token response rewards.
- The authors theoretically justify their method and point out its convergence to the same global optimum as DAPO, which motivates its generalizability to other architectures.
- The method is evaluated on models of fairly variable sizes, and the authors conduct ablation studies on sensitive hyperparameters.
- The authors' method shows significant improvement on the mathematical problems presented in Table 1.

**Weaknesses:**
- In the Discussion section, the authors mention that they are comparing entropies within a batch of problems, which is confusing, as the formulas previously did not state the relationship between the various elements of the minibatch during training. It would be nice if the authors clarified this point.
- Baselines are insufficient for comparison. The authors presented a comparison with GRPO and DAPO, which is logical given their theoretical connection to the method proposed by the authors. However, it would be useful to see a comparison with newer methods like GMPO and GSPO.
- The authors do not include pass@1 in Table 1, which is curious and not entirely trivial. It would be good to hear the authors' rationale for this measurement system.

---

> ### Author Rebuttal · Authors · 2026-03-31
>
> We thank the reviewer for carefully reading and providing constructive suggestions. Below we address each weakness/question in turn.
>
> ---
>
> **Weakness #1 – Clarification on within-batch entropy comparison**
> - **Formula Level (Per-Group)**: Entropy normalization (e.g., $\frac{H_{i,t}}{\sum_{k=1}^{n}H_{k,t}}$) is computed strictly within a single group of responses for a specific prompt. We do not normalize across different prompts.
> - **Training Dynamics (Per-Batch)**: The “batch-level comparison” in Discussion refers to the implicit curriculum effect: a batch aggregates groups of varying difficulties, so the model produces larger gradient signals for solvable but high-entropy problems. This focuses learning on the frontier of the model’s capabilities without cross-prompt normalization.
>
> We will revise the Discussion to explicitly distinguish per-group normalization from per-batch implicit dynamics.
>
> ---
>
> **Weakness #2 – Insufficient experimental analysis**
> We have added a comparison with GSPO below. Additional experiments are underway and will be included in a future version.
>
> |                        | AIME 2024 | AIME 2025 | MATH 500 |
> |------------------------|--------|--------|----------|
> | Qwen2.5-7b base        | 16.67  | 8.33   | 50.02    |
> | GSPO                   | 33.67  | 16.33  | **80.20** |
> | GTPO                   | **34.33** | **16.67** | 80.16    |
>
> ---
>
> **Weakness #3 – Pass@1 measurement clarification**
> Thank you for your comment. Our Mean@32 is exactly pass@1 (we used that term to emphasize the metric). We will unify the notation in the revised version.
>
> ---
>
> **Question #1 – Entropy collapse in GRPO and training dynamics**
> Entropy collapse in standard GRPO was one of the motivations for our work. Appendix E.1 (Figure 6) shows entropy and response length dynamics. Both GTPO and GRPO-S induce an “entropy rebound” (avoiding collapse), which correlates with longer responses and improved the model's reasoning capabilities. This confirms that dynamic reward redistribution prevents entropy collapse and sustains exploration.
>
> ---
>
> **Question #2 – Balance of positive/negative responses and convergence**
>
> In GRPO-style training, the core mechanism is rollout (self-sampling), which introduces stochasticity. The probabilities of positive and negative responses naturally evolve as the model learns, and this is largely uncontrollable within the RL framework. However, virtually all data contain both outcomes, and the contrast is the main driving force for RL learning.
>
> Convergence is proved in Theorem 2.4 and Appendices B.3 and B.4.
>
> Importantly, our dynamic reward redistribution ensures that even if a batch is all‑correct or all‑incorrect, the advantage function remains non‑zero, providing a gradient step and utilizing those samples.
>
> ---
>
> **Question #3 – Inverse entropy vs. negative entropy and training dynamics**
>
> We sincerely thank the reviewer for this excellent question regarding the choice of inverse entropy over negative entropy. We interpret your suggestion of "negative entropy" to mean $-H$. (However, if you envisioned a formulation like $ H_{max} - H $, we completely agree that it is a highly elegant alternative! We plan to explore and compare its empirical performance against our current method in future work). Assuming the $-H$ interpretation, we justify our choice of inverse entropy through two main aspects: our core design intent and the mathematical stability of the resulting weights.
>
> - **Design Intention: Penalizing Confident Errors** Our reward reshaping for negative responses is explicitly designed to disproportionately penalize "high-confidence errors." We want the model to learn that being confidently wrong (low $H$) is much worse than being hesitantly wrong (high $H$). To achieve this, we aggregate the base penalties -1 across all incorrect responses at timestep $t$ and redistribute them using an entropy-derived weight. To assign a larger absolute penalty to a low-entropy token, that token must hold a larger relative weight. The reciprocal, $1/H$, naturally achieves this: as $H$ decreases, $1/H$ increases, capturing a larger share of the negative reward mass. Conversely, if we were to use $-H$, a low-entropy token would receive a smaller relative weight, resulting in a smaller absolute penalty. This would directly contradict our core design objective.
> - **Mathematical Properties and Numerical Stability** You raise a very valid practical concern regarding numerical instability when dividing by a near-zero entropy value. However, because $1/H$ appears in both the numerator and the denominator to form a normalized ratio, the term remains mathematically well-behaved. Specifically, even as $H \to 0$ and $1/H \to \infty$, the normalized weight $\frac{1/H}{\sum (1/H)}$ strictly converges to a bounded, well-defined constant (e.g., $1$ if a token has the uniquely lowest entropy, or $1/k$ if $k$ tokens tie for the minimum).

---

> > ### Author Rebuttal · Reviewer_tZj3 · 2026-04-03
> >
> > All my questions were correctly responded by the authors. Therefore, I will maintain my current positive score.

---

> > > ### Author Response · Authors · 2026-04-08
> > >
> > > Thank you very much for taking the time to review our rebuttal. We are especially grateful that our rebuttal has adequately addressed your concerns, which means a great deal to us.
> > >
> > > We sincerely apologize for not being able to express our gratitude earlier. We had intended to respond sooner, but encountered an issue with the submission system and were only able to successfully submit our response today.
> > >
> > > We truly appreciate your positive assessment of our work, including the importance of addressing reward shaping in GRPO-like methods and introducing a value-function-like perspective, the novel use of model entropy as a confidence signal for per-token rewards, the theoretical guarantees on convergence and their implications for generalizability, as well as the comprehensive empirical evaluation across diverse model scales, detailed ablations, and strong performance on mathematical benchmarks. We are grateful for your recognition of these aspects.
> > >
> > > We thank you again for your time and consideration.

---

### Official Review · Reviewer_BtpL · 2026-03-10

**Soundness:** 3
**Presentation:** 3
**Significance:** 2
**Originality:** 2
**Overall Recommendation:** 4
**Confidence:** 4

**Summary:**

The authors propose **GTPO and GRPO-S**, which introduce **token-level fine-grained rewards** to address the limitations of sequence-level policy gradient methods.
The proposed approach adjusts the strength of token-level policy gradients based on the **entropy of the policy at each token**.
The method demonstrates strong performance on **AIME** and **MATH 500** benchmarks.

**Compliance With Llm Reviewing Policy:**

Affirmed.

**Final Justification:**

This paper is experimentally sound and makes a significant contribution. I am leaning toward accepting it.

**Key Questions For Authors:**

## Questions

- **Comparison with other fine-grained reward methods**

  How does the proposed method differ from other fine-grained reward approaches such as **PRM (Process Reward Models)**?
  It would be interesting to analyze the **distributional differences between PRM-based rewards and entropy-based weighting**.

- **Relation to entropy maximization in RL**

  The proposed entropy-based reward redistribution appears related to the **entropy maximization objective** commonly used in reinforcement learning.
  In **GRPO**, higher entropy is partially encouraged through the **KL-divergence constraint with the reference model**.

  How is the **KL-divergence with the reference model handled in GTPO or GRPO-S**?
  Additionally, how does the **increase in entropy affect the learning dynamics** in practice?

**Limitations:**

yes

**Strengths And Weaknesses:**

## Strengths
----
- **Novel reward redistribution idea**

  The idea of redistributing rewards at the **token level rather than the sequence level** is interesting.
  Recently, there has been growing interest in training methods that place greater emphasis on **low-entropy tokens**, and this work extends that idea within the **GRPO framework**, broadening the discussion in this research direction.

- **Clear writing and presentation**

  The paper is well written and easy to understand.

- **Empirical validation**

  The experimental results support the authors' claims.
----
## Weaknesses
----
  - Since entropy must be computed and normalized for **every token**, the method may introduce additional computational cost.
    However, this does not appear to be a critical drawback given the performance improvements.

  - The method may be **sensitive to hyperparameters**.
    In particular, entropy values may vary across models, which may require careful hyperparameter tuning.

---

> ### Author Rebuttal · Authors · 2026-03-31
>
> We thank the reviewer for carefully reading and providing constructive suggestions. Below we address each weakness/question in turn.
>
> ---
>
> **Weakness #1 – Computational cost consideration**
> *We sincerely thank you for your thorough review and for recognizing the value of our performance gains. We want to take this opportunity to gently clarify that our method actually introduces virtually **zero additional computational overhead**.*
>
> - **Computing Logits**: This is an inherent, unavoidable step in all baseline algorithms and requires no extra computation.
> - **Computing Entropy from Logits**: This mathematical operation is fully parallelizable across the entire sequence. Calculating the entropy for all tokens simultaneously takes the exact same time as computing it for a single token, adding a negligible $O(V)$ complexity (where $V$ is the vocabulary size).
> - **Arithmetic Averaging**: Averaging these token-level entropies across the complete response adds no meaningful time complexity.
>
> Therefore, our framework seamlessly integrates into existing training pipelines without increasing the overall computational cost or training time.
>
> ---
>
> **Weakness #2 – Considerations for hyperparameter sensitivity**
> *We sincerely thank the reviewer for this practical insight. You are absolutely right that intrinsic entropy naturally varies across different base models, which invariably requires hyperparameter tuning.*
>
> However, we would like to gently point out that hyperparameter sensitivity is a universal characteristic of RL-based alignment algorithms, rather than a vulnerability unique to our method. For instance, the robust DAPO baseline itself relies on meticulously tuned entropy clipping thresholds (e.g., $\epsilon_{high}=0.28$) to maintain training stability.
>
> To make our framework straightforward to deploy, we have established reliable empirical bounds that significantly reduce the need for exhaustive tuning. In our experiments (e.g., with Qwen2.5-7B), we observed that the token-level GTPO is relatively more sensitive to the perturbation weight ($\alpha_2$). While a higher $\alpha_2/\alpha_1$ ratio yields more significant performance gains, it should generally not exceed $0.2$. For the sequence-level GRPO-S, the $\beta_2/\beta_1$ ratio operates best when kept below $0.12.$ While these exact upper bounds will naturally scale depending on the specific base model, these established heuristics provide a highly stable and transferable starting point for future practitioners.
>
> ---
>
> **Question #1 – Comparison with PRM**
> *Thank you for your valuable comments. Here we clarify the differences between our method and PRM.*
>
> Our current approach assigns rewards by redistributing them according to a set of macro-level criteria defined by the model itself. It provides a fair reward mechanism for all tokens, including those involved in the reasoning process. In this sense, our method shares certain similarities with PRM.
>
> However, PRM typically relies on explicit intermediate reward signals, which often depend on a well-trained reward model or human annotations. In contrast, our method achieves similar fine-grained, entropy-based rewards through an implicit mechanism, leveraging endogenous signals from the model itself. This distinguishes our approach from conventional PRM.
>
> That said, we would like to note that our method is structurally compatible with PRM, and we are currently experimenting with their combination.
>
> ---
>
> **Question #2 – Relation to entropy maximization in RL**
> *We thank the reviewer for highlighting this important connection. We agree that contrasting our approach with traditional entropy maximization provides valuable clarity.*
>
> - **Global Regularization vs. Dynamic Redistribution**:  Traditional RL treats entropy as a global, additive regularization term to maintain overall policy randomness. In contrast, our framework repurposes entropy for active reward shaping. We do not simply add an entropy penalty; we use it as a relative weight to dynamically redistribute the reward signal. This transforms entropy from a blunt global parameter into a fine-grained, localized signal that directs the model to maintain uncertainty specifically at critical decision points, avoiding local optima.
> - **Handling of KL-Divergence**:  We handle the KL-divergence penalty against the reference model exactly as it is faithfully implemented in the DAPO baseline.
> - **Practical Learning Dynamics ("Entropy Rebound")**:  The practical impact of this design is empirically demonstrated in Appendix E.1 (Figure 6). While the baseline DAPO quickly suffers from policy collapse—where entropy continuously drops into a narrow, deterministic state—our method induces a distinct "entropy rebound". This sustained exploration directly correlates with increased average response lengths, indicating that the model is actively engaging in longer, deeper, and ultimately more accurate CoT reasoning.

---

> > ### Author Rebuttal · Reviewer_BtpL · 2026-03-31
> >
> > Thank you for your response. My concerns have been addressed, and I will maintain the current score.

---

> > > ### Author Response · Authors · 2026-04-04
> > >
> > > Thank you very much for taking the time to review our rebuttal. We are especially grateful that our rebuttal has adequately addressed your concerns, which means a great deal to us.
> > >
> > > We truly appreciate your positive assessment of our work, including the novelty of our reward redistribution idea at the token level, its connection to and extension of recent efforts on entropy-aware training within the GRPO framework, the clarity and accessibility of our presentation, as well as the empirical results supporting our claims. We are grateful for your recognition of these aspects.
> > >
> > > We thank you again for your time and consideration.

---

### Official Review · Reviewer_jy1Z · 2026-03-14

**Soundness:** 3
**Presentation:** 3
**Significance:** 3
**Originality:** 3
**Overall Recommendation:** 4
**Confidence:** 3

**Summary:**

This paper addresses the coarse-grained credit assignment problem in GRPO and DAPO by proposing a Dynamic Entropy Weighting mechanism that reshapes the reward signal into fine-grained rewards before it is used by the policy model. It rewards high-entropy tokens in correct trajectories and penalizes low-entropy tokens in incorrect trajectories, and further proposes GTPO and GRPO-S to realize token-level reward shaping and sequence-level reward shaping. Experiments on multiple mathematical benchmarks demonstrate the effectiveness of the proposed method.

**Compliance With Llm Reviewing Policy:**

Affirmed.

**Key Questions For Authors:**

See weaknesses

**Limitations:**

yes

**Strengths And Weaknesses:**

### Strengths

1. The paper has a clear motivation, and the coarse-grained credit assignment problem in GRPO and DAPO is important.
2. The proposed method is easy to follow and cover both token-level and sequence-level formulations.
3. The paper provides a relatively systematic analysis of the motivation and reasonableness of the method design, which supports part of its claims.
4. The experimental results consistently show better performance than the baselines, and the analysis of training dynamics is convincing.


### Weaknesses

1. The paper proves raw positive reward mass conservation, but the actual training objective uses the separately normalized advantages in Eq.(6). It is still unclear how this can guarantee that the optimization direction or the global optimum remains unchanged.
2. The motivation for assigning larger penalties to low-entropy tokens is not clear enough. Low-entropy tokens may also be syntactically certain tokens, but not the parts that lead to error reasoning. The authors should provide some experimental or theoretical evidence to support this motivation.
3. The experimental analysis is still insufficient. The evaluation is only conducted on math reasoning tasks, without evaluating OOD reasoning ability, and there is no ablation on the positive entropy reward and the inverse entropy penalty on negative samples.

---

> ### Author Rebuttal · Authors · 2026-03-31
>
> We thank the reviewer for carefully reading and providing constructive suggestions. Below we address each weakness in turn.
>
> ---
>
> **Weakness #1 – Why is the global optimum unchanged under separate normalization?**
> The theoretical guarantees for unsuccessful sequences are perfectly symmetric to the positive part.
>
> - **Conservation of negative reward mass**: For active failed sequences at step $t$ (with $\alpha_1+\alpha_2=1$), we have $\sum_{j\in O_t^{-}} \tilde{r}_{j,t}^{-} = -|O_t^{-}|$. The negative reward mass is strictly redistributed without net bias.
> - **Asymptotic consistency**: The deviation from the standard penalty is proportional to inverse entropy $1/H$. As the policy converges, entropy variation among failed sequences vanishes ($\lim_{k\to\infty} (1/H_{p,t})/(1/H_{q,t})=1$), so the pointwise deviation approaches zero.
> - **Shared global optimum**: Because both positive and negative shapings conserve total reward masses, GTPO performs a zero-sum redistribution of the DAPO objective. The overall gradient decomposes as
>   $\nabla J_{\text{GTPO}} = \nabla J_{\text{DAPO}} + \mathbb{E}\big[\text{Cov}(\Delta_{\text{entropy}}, \nabla\log\pi_\theta)\big].$
>   This covariance term vanishes at optimality (when entropy variation disappears). Hence the global optimum remains mathematically identical.
>
> ---
>
> **Weakness #2 – Motivation for penalizing low‑entropy tokens.**
> - **Motivation – penalizing “stubborn” errors**: When a model generates an incorrect proof with high confidence (low entropy), it is trapped in a flawed reasoning path. To break this cognitive trap and force exploration, we aggressively penalize these confident mistakes – but only through relative uncertainty redistribution, not in absolute vacuum.
> - **Mechanism for low-entropy syntactic tokens (e.g., “Therefore”, “Let”)**: These naturally have low entropy but do not cause reasoning errors. In the DAPO baseline, all tokens in a failed response receive a uniform $-1$ penalty. GTPO redistributes these $-1$ penalties using relative entropy comparisons. Because syntactically certain tokens have *uniformly* low entropy across sequences, their relative differences are negligible; their redistributed penalty remains very close to the baseline $-1$. Thus they are not disproportionately punished; the dynamic shifting concentrates only on critical reasoning junctures where entropy varies significantly.
> - **Related Work**: The following works are similar in spirit to our approach.
>
> [1] ICPO: Intrinsic Confidence-Driven Group Relative Preference Optimization for Efficient Reinforcement Learning
>
> [2] Honesty over Accuracy: Trustworthy Language Models through Reinforced Hesitation
>
> [3] GEM: Generative Entropy-Guided Preference Modeling for Few-Shot Alignment of LLMs
>
> ---
>
> **Weakness #3 – Insufficient experimental analysis.**
> - **OOD reasoning evaluation**: We fully agree. To evaluate the performance of GTPO and GRPO-S on OOD tasks, we tested the trained models on three commonly used OOD scientific reasoning benchmarks: MMLU-Pro [1], GPQA-Diamond [2], and ARC-Challenge [3]. We also include a comparison with the GSPO method. Experimental results show that both GTPO and GRPO-S still maintain a clear advantage on OOD reasoning tasks.
>
> |                        | MMLU-Pro [1] | GPQA-Diamond [2] | ARC-Challenge [3] |
> |------------------------|--------------|----------|--------------------|
> | qwen2.5-base           | 35.42        | 40.91    | 71.19              |
> | GSPO                   | 39.28        | 48.83    | 83.65              |
> | GTPO                   | **40.34**    | **50.13**| **84.08**          |
> | GRPO-S                 | 39.46        | 49.38    | 83.19              |
>
> [1] Mmlu-pro: A more robust and challenging multi-task language understanding benchmark
>
> [2] Gpqa: A graduate-level google-proof q&a benchmark
>
> [3] Think you have Solved Question Answering? Try ARC, the AI2 Reasoning Challenge
>
> - **Ablation on positive vs. negative reward shaping**: We respectfully note a slight misunderstanding. GTPO’s positive entropy reward and negative inverse entropy penalty are *symmetric halves of a single unified design principle* – amplifying entropy signals to enforce dynamic exploration. They are not modular “add‑ons”.
>   - Positive samples: encourage exploration of better parameter spaces when the answer is correct but accuracy is suboptimal.
>   - Negative samples: force the model to break out of “stubborn” reasoning paths when it is confidently wrong.
>   Joint operation is strictly required for robustness. For example, if a batch happens to be all‑correct or all‑incorrect, standard GRPO gradients collapse to zero; our joint design ensures that relative entropy differences still provide a non‑zero gradient step. Ablating either half would break the zero‑sum conservation properties and the unified logic of the algorithm.

---

> > ### Author Rebuttal · Reviewer_jy1Z · 2026-04-03
> >
> > Thank you for your response. I will keep my score.

---

> > > ### Author Response · Authors · 2026-04-04
> > >
> > > Thank you very much for taking the time to review our rebuttal. We are especially grateful that our rebuttal has adequately addressed your concerns, which means a great deal to us.
> > >
> > > We truly appreciate your positive assessment of our work, including the clear motivation and importance of the coarse-grained credit assignment problem, the intuitive and comprehensive formulation covering both token- and sequence-level perspectives, the systematic analysis supporting the method design, as well as the strong empirical performance and convincing training dynamics compared to baselines.
> > >
> > > We sincerely thank you again for your positive assessment and valuable feedback.

---

### Decision · Program_Chairs · 2026-04-30

**Decision:**

Accept (regular)

**Comment:**

This paper introduces GTPO and GRPO-S, novel frameworks addressing the coarse-grained credit assignment problem in RL. Reviewers find the motivations clear, appreciate systematic analysis of training dynamics, and think performance results are strong. I encourage the authors to fully address reviewer' questions and concerns in the final version.